# THE DEVIL IS IN THE OBJECT BOUNDARY: TOWARDS ANNOTATION-FREE INSTANCE SEGMENTATION USING FOUNDATION MODELS

**Cheng Shi & Sibei Yang** [†]
School of Information Science and Technology
ShanghaiTech University
`{shicheng2022,yangsb}@shanghaitech.edu.cn`

## ABSTRACT

Foundation models, pre-trained on a large amount of data have demonstrated impressive zero-shot capabilities in various downstream tasks. However, in object detection and instance segmentation, two fundamental computer vision tasks heavily reliant on extensive human annotations, foundation models such as SAM and DINO struggle to achieve satisfactory performance. In this study, we reveal that the devil is in the object boundary, *i.e.*, these foundation models fail to discern boundaries between individual objects. For the first time, we probe that CLIP, which has never accessed any instance-level annotations, can provide a highly beneficial and strong instance-level boundary prior in the clustering results of its particular intermediate layer. Following this surprising observation, we propose *Zip* which **Z**ips up CL**ip** and SAM in a novel classification-first-then-discovery pipeline, enabling annotation-free, complex-scene-capable, open-vocabulary object detection and instance segmentation. Our Zip significantly boosts SAM's mask AP on COCO dataset by 12.5% and establishes state-of-the-art performance in various settings, including training-free, self-training, and label-efficient finetuning. Furthermore, annotation-free Zip even achieves comparable performance to the best-performing open-vocabulary object detectors using base annotations. Code is released at https://github.com/ChengShiest/Zip-Your-CLIP

## 1 INTRODUCTION

The primary objective of both *object detection* and *instance segmentation* is to localize and classify objects within an image. Although significant progress has been made in them, the costly, challenging, and sometimes even infeasible annotation collection limits fully-supervised approaches (Carion et al., 2020; He et al., 2017; Ren et al., 2015; Girshick, 2015). Recently, a wave of vision foundation models, such as DINO (Caron et al., 2021; Oquab et al., 2023), CILP (Radford et al., 2021), and SAM (Kirillov et al., 2023), has emerged and demonstrated impressive zero-shot generalization capabilities at various downstream tasks (Wang et al., 2023; Zhou et al., 2022c; Kirillov et al., 2023). A straightforward idea is to directly transfer and synergistically utilize foundation models for object detection and instance segmentation, thereby eliminating the need for additional human annotations.

In this work, we study ***annotation-free*** object detection and instance segmentation, where the item "annotation-free" represents without needing any instance annotations and in-domain human labels. We aim to directly generalize foundational models in a zero-shot manner to achieve object detection and instance segmentation. To attain this objective, the discovery and precise localization of object proposals are indispensable. Given that DINO has been employed for unsupervised object discovery in object-centric images from ImageNet (Deng et al., 2009), and SAM is capable of generating valid segmentation masks for any things and stuff, we initially endeavor to harness them to localize multiple objects on more complex scenes from COCO (Lin et al., 2014):

---

[†]Corresponding author

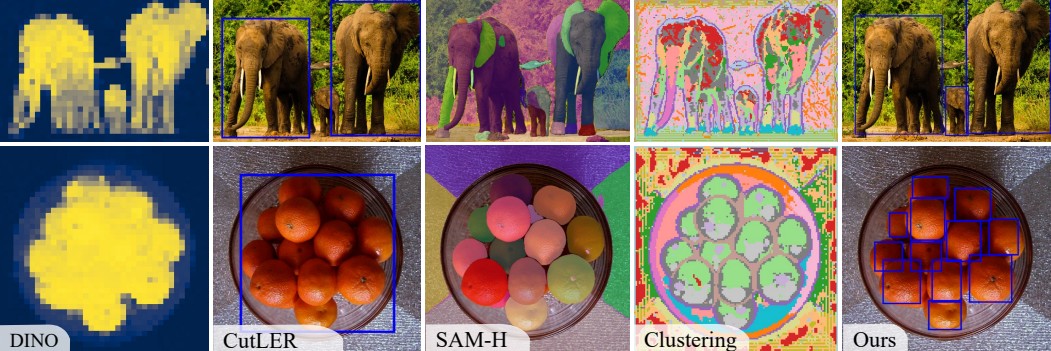

Figure 1: **Comparison of different annotation-free object discovery methods and zero-shot SAM-H.** Previous state-of-the-art methods rely on DINO, struggle to discern between instance-level objects, and often miss potential objects within the background. On the other hand, SAM generates valid segmentation masks but struggles to determine the confidence of a particular mask representing an object. In contrast, our approach, employing clustering outcomes of CLIP's intermediate features, can effectively outline boundaries between objects to differentiate them.

- Object discovery relying upon the foreground object mask from DINO struggles to discern between different object instances, such as different oranges in Figure 1, resulting in only being able to localize salient regions in simple scenes instead of instance-level objects.

- SAM produces numerous valid segmentation masks, including some object instances, achieving superior recall rates compared to DINO in object localization. However, SAM still yields relatively low precision due to its semantic-unaware and mostly edge-oriented approach, making it hard to discern the confidence of a particular mask representing an object. We observe that SAM tends to assign higher confidence scores to smoother "planes", as shown in Appendix B. Furthermore, considering CLIP enables the recognition of image regions, we combine the CLIP classification score with the SAM localization score using a geometric mean. This enhancement elevates the confidence scores of masks, resulting in a marginal increase in precision, though it falls short of complete satisfaction.

At its essence, both DINO and SAM are limited to distinguish objects, that is, ***to discern the boundaries between objects.*** DINO cannot detect the edges between similar and closely positioned objects, whereas SAM can clearly delineate edges but cannot distinguish which ones are object boundaries of interest and which are part of the objects' interiors. For instance, in the case of an elephant's mask, there may be distinct edges between head and ear, but these edges do not serve as boundaries distinguishing the elephant from other objects. The devil is in the object boundaries!

To overcome the challenge, we probe the CLIP's ability to discern the boundaries between objects instead of utilizing DINO or SAM to discover object boundaries. Our key insights are: 1) CLIP's features encompass the rich semantics learned from a vast corpus of image-text pairs, and semantics are a necessary condition for defining what constitutes an object boundary. For instance, in the case of "elephant", edges between "nose" and "head" are not considered object boundaries, but they are so for "nose" itself. 2) Although CLIP is trained on image-level alignment with text, it may have encountered fine-grained image-text pairs describing multiple local objects and their relations during training, thereby enabling it to identify and distinguish objects. After encountering several unsuccessful attempts, we observe that ***clustering the features of CLIP's specific middle layer (as shown in Figure 1) can delineate the boundaries between individual objects!*** The visualization results clearly outline the objects in various complex scenes, such as three elephants and clusters of oranges. These widely observed clustering results encourage our study. To the best of our knowledge, we are the first to discover CLIP's ability to clearly outline object boundaries. In contrast, some semantic segmentation methods (Zhou et al., 2022a; Li et al., 2023) rely on clustering CLIP's features to extract regions with the same semantics but cannot distinguish objects. Then, we propose a specific boundary score metric to extract object boundaries and combine with semantics to detect multiple objects in Figure 3, as detailed in Section 3.4. ***Notably, now SAM can be effectively utilized to generate instance-level segmentation masks!*** Indeed, the issue of SAM's inability to distinguish whether edges belong to object boundaries or the inner edges of objects has been overcome by CLIP's

ability to outline object boundaries. For the first time, we achieved the zero-shot generalization of CLIP and SAM foundation models to instance-level object localization and segmentation.

Furthermore, one incidental benefit of using CLIP to discover objects is to subsequently employ it for classifying the discovered object proposals, achieving open-vocabulary instance segmentation. Unfortunately, the image-level alignment of CLIP makes it susceptible to contexts when classifying proposals, resulting in objects being classified with low confidence and even misclassified. In Figure 8 of Appendix F, CLIP even misclassifies the salient object "car" as a "cat" in such a simple scene .

Therefore, we propose a ***classification-first-then-discovery pipeline*** rather than the conventionally used discovery-followed-by-classification one. As shown in Figure 3, for a given open-vocabulary category, we first extract the class-specific semantic clues in the image by computing the similarity between the category and the features of CLIP's final layer. Notably, the semantic clues capture per-pixel alignment instead of per-image, which can avoid the previously mentioned mismatch issue between the CLIP's scene-centric training and our object-centric requirement, despite only capturing relatively rough regions. Then, we detect objects with precise boundaries by first clustering on features of CLIP's middle layer to extract fragments and then performing fragment selection to form objects according to the boundary score metric and the semantic clues. Finally, SAM can be prompted by our individual instance-level object mask, leading to a more accurate mask.

Our contributions are multi-fold: **1)** We discover that the devil is in the object boundaries when employing foundation models such as DINO and SAM for annotation-free object detection and instance segmentation. Our study is the first to discover the potential of CLIP's particular middle layer in outlining object boundaries to overcome the challenge. **2)** We propose a novel classification-first-then-discovery framework, namely ***Zip***, enabling annotation-free, complex-scene-capable, open-vocabulary object detection and instance segmentation. Our ***Zip*** pipeline is the first to effectively *z*ip up CL***IP*** and SAM foundation models to segment objects in an annotation-free manner, according to object boundaries and semantic clues. **3)** Zip in a training-free manner, *i.e.*, without the need for any additional training and by directly leveraging off-the-shelf foundation models, exhibits strong zero-shot performance across various evaluation settings, significantly improving class-agnostic (+8.7% AP), class-aware (+8.8% AP), and comparable performance on open-vocabulary object detection without any base annotation. **4)** Zip, by combining training strategies such as self-training and data-efficient fine-tuning (with 5% data), can further improve AP to 18.2% and 28.1%. Zip with self-training significantly outperforms state-of-the-art annotation-free instance segmenter by 14.4% AP50 under the same settings.

## 2 RELATED WORK

**Visual Foundation Models.** Without the need for finetuning, the foundation models (Brown et al., 2020; Ouyang et al., 2020; Kirillov et al., 2023; Radford et al., 2021; Li et al., 2021; Caron et al., 2021; Oquab et al., 2023; Chowdhery et al., 2022), pre-trained on carefully designed tasks have demonstrated impressive zero-shot capabilities for various downstream tasks. The recently introduced DINO-v2 (Oquab et al., 2023), pre-trained in a self-supervised manner on a diverse range of curated data sources, shows astounding performance across various downstream tasks. The contrastive vision-language models (VLMs) such as CLIP (Radford et al., 2021) and ALIGN (Li et al., 2021), train a dual-modality encoder on large-scale image-text pairs to learn transferable visual representations under text supervision using contrastive loss. The transferable visual representations bolster a myriad of downstream tasks, including image classification (Zhou et al., 2022c;b; Shi & Yang, 2023b), image synthesis (Rombach et al., 2021; Huang et al., 2024), video comprehension (Lin et al., 2022; Tang et al., 2023b), semantic segmentation (Zhou et al., 2022a; Li et al., 2023), referring segmentation Dai & Yang (2024); Tang et al. (2023a), and object detection (Zhou et al., 2022d; Feng et al., 2022; Gu et al., 2022; Shi & Yang, 2023a). SAM (Kirillov et al., 2023) introduces the pioneering foundation model for image segmentation, pre-trained using one billion masks and designed to respond to diverse input prompts such as points, bounding boxes, masks, and text. In standard object detection, the potential of the foundation model has yet to be validated. Our goal is to leverage the foundation model to explore its performance in object detection and instance segmentation tasks.

**Unsupervised Object Detection and Instance Segmentation.** As fundamental tasks in computer vision, object detection and instance segmentation require extensive human annotation to yield good performance (He et al., 2017; Carion et al., 2020). Many previous works (Wang et al., 2022b; 2023;

Siméoni et al., 2021; Wang et al., 2022a; Van Gansbeke et al., 2022) have explored unsupervised approaches to circumvent this bottleneck. DINO (Caron et al., 2021) first unveiled that the self-attention modules of the terminal block in a self-supervised vision (Dosovitskiy et al., 2020) will automatically learn to segment the foreground objects. Following works (Wang et al., 2022b; 2023; Siméoni et al., 2021) build upon DINO's observation and utilize DINO's patch features to construct various similarity matrices for separating a single foreground object in each image. To detect multiple objects, CutLER (Wang et al., 2023) proposes MaskCut, which masks out the similarity matrix using the foreground mask, and then iteratively separates the objects. However, as shown in Figure 1, foreground objects do not inherently contain instance information. Therefore, these previous methods fail to detect individual instances when multiple objects are densely clustered together, and they also miss inconspicuous objects in the background. Simultaneously, due to the lack of semantic understanding in pre-trained networks, they can only localize objects without being able to classify them. In contrast, our work not only leverages CLIP (Radford et al., 2021) to discover objects based on our observations that the intermediate output of CLIP offers a strong prior for object discovery but also to classify objects based on the image-text alignment from the CLIP pre-training.

## 3 METHOD

In this section, we first revisit the CLIP model and introduce how to extract dense semantic clues from it in Section 3.1. Then, we introduce our new discovery that clustering the features of CLIP's particular middle layer can delineate the boundaries between objects (Section 3.2). To synergistically leverage CLIP's property for object boundary discovery and SAM's ability to generate segmentation masks, we propose a novel classification-first-then-discovery Zip framework, enabling annotation-free object detection and instance segmentation (Section 3.3). In Section 3.4, we provide details on how Zip derives the localization of individual objects from CLIP's clustering results.

### 3.1 PRELIMINARY

**A Revisit of Contrastive Language-Image Pre-Training.** CLIP (Radford et al., 2021) is trained on a vast corpus of image-text pairs using the contrastive learning approach. It encodes the input image $\mathcal{I}$ and input text $\mathcal{T}$ by an image encoder $\text{Enc}_I(\cdot)$ and a text encoder $\text{Enc}_T(\cdot)$, respectively. To compute the similarity between the image and text, CLIP employs the following formula:

$$\mathcal{S}_{\text{CLIP}}^{\text{img}}(\mathcal{I}, \mathcal{T}) = \cos(\text{Enc}_I(\mathcal{I}), \text{Enc}_T(\mathcal{T})), \tag{1}$$

where $\cos(\cdot, \cdot)$ denotes the cosine similarity.

**Extract Dense Semantic Clues from CLIP.** As CLIP is primarily intended for image-level predictions, it is not readily feasible to extract local patch predictions (*i.e.*, similarities between image patches $\mathcal{I}_{patch}$ and text $\mathcal{T}$) from CLIP. Previous works (Zhou et al., 2022a; Li et al., 2023) propose a specific modification in the last attention-pooling layer of $\text{Enc}_I(\cdot)$ to achieve the local predictions. Given an image $\mathcal{I} \in \mathbb{R}^{h \times w \times 3}$ with the size of $h \times w$ and its patch features obtained by passing the image through the $\text{Enc}_I(\cdot)$ except for the last layer, the similarities between the patches $\mathcal{I}_{patch}$ and the text $\mathcal{T}$ are computed as follows,

$$\mathcal{S}_{\text{CLIP}}^{\text{patch}}(\mathcal{I}_{\text{patch}}, \mathcal{T}) = \cos(\text{Proj}(\hat{\text{Enc}}_I(\mathcal{I}_{\text{patch}})), \text{Enc}_T(\mathcal{T})), \tag{2}$$

where $\text{Proj}(\cdot)$ is the value-embedding projection of the last attention-pooling layer and $\hat{\text{Enc}}_I(\cdot)$ is the encoder $\text{Enc}_I(\cdot)$ except for this pooling layer. When a class name like "elephant" is given as the input text, the $\mathcal{S}_{\text{CLIP}}^{\text{patch}}(\cdot, \cdot)$ function allows us to calculate similarities between image patches and the class, and thus obtain to an activation map indicating rough regions for that class in the image. One resulting region is visualized in "Semantic Clues" of Figure 3.

### 3.2 PROBING CLIP FOR DISCERNING OBJECT BOUNDARY

In this section, we probe a new property of CLIP, which is the discovery of boundaries between objects. Figure 2 presents the results of clustering using features from the particular middle layers of CLIP (conv4_x layer). As depicted, under various challenging scenarios, CLIP can distinctly outline the edges of objects of interest, represented by the gray cluster in Figure 2B and the orange cluster

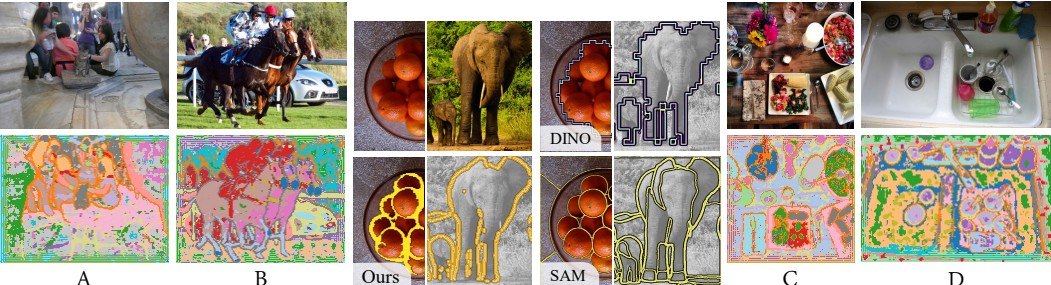

Figure 2: **Clustering result and boundary comparison between different methods.** In our clustering results from A to D, the boundaries between objects are successfully clustered into particular clusters marked in orange or gray, serving as strong priors for segmenting objects of the same category. In the middle, we showcase the edges extracted by various methods. DINO provides only prominent outer edges, while SAM segments all edges, but such edges cannot distinguish which ones are object boundaries of interest. The devil is in the boundary.

in other cases. These clusters of edges define the boundaries between objects ("Ours" in Figure 2), enabling the localization of each individual object. In contrast, DINO can only obtain the outer edges of the salient regions, while SAM captures all prominent edges. However, unlike ours, the edges extracted by both SAM and DINO do not directly correspond to boundaries between instance-level objects. Notably, this observation is universal, and these four cases from A to D are randomly selected for visualization. For a comprehensive analysis of this general observation, please see Appendix C.1. Specifically, we employ standard K-Means clustering to obtain such impressive results. Please refer to Appendix C for details.

## 3.3    PROBLEM STATEMENT AND ZIP FRAMEWORK

In this section, we first formulate the problem statement. Then, we introduce our classification-first-then-discovery pipeline, which is the first to effectively and synergistically utilize CLIP and SAM foundation models to detect and segment objects in an annotation-free manner, according to semantic clues (Equ. 2) and object boundaries (Sec. 3.2).

**Annotation-Free Object Detection and Instance Segmentation** seeks to detect and segment objects in an image belonging to a set of object categories $\mathcal{T}_{\text{class}}$ without instance annotations and in-domain human labels. More specifically, given an image $\mathcal{I} \in \mathbb{R}^{h \times w \times 3}$, for each category with a text name $\mathcal{T} \in \mathcal{T}_{\text{class}}$, the segmenter is required to predict all the instances belongs to the category $\mathcal{T}$ and their segmentation masks. Considering the flexibility and adaptability of the **open-vocabulary** textual category, this task can facilely extend to various object detection and instance segmentation tasks. In this section, we introduce our instance segmentation method to simplify the demonstration because instance masks can be converted into bounding boxes of objects.

**Our Zip pipeline, which follows a classification-first-then-discovery approach**, consists of four steps, as illustrated in Figure 3.

- Classification first to obtain semantic clue prior. For each textual category $\mathcal{T}$, we first obtain an image-level similarity $S_{\text{T}}^{\text{img}}$ and a patch-level activation map $S_{\text{T}}$ via Equ. 1 and Equ. 2, respectively. Considering that the semantic activation map $S_{\text{T}}$ ("Semantic Clues" in Figure 3) provides only rough object regions and lacks the ability to distinguish clustered objects of the same class, we depart from the CLIP+SAM  (Li et al., 2023) of directly selecting points from the activation map to prompt SAM for instance segmentation. Instead, we leverage the semantic activation map as a strong prior in two distinct ways: 1) To assist CLIP in obtaining stable clustering results for outlining object boundaries, as discussed in Section 3.2. 2) To identify the group of clustered fragments belonging to the same object, thereby enabling the localization of bounding boxes for each individual object, will be detailed in Section 3.4.

- Clustering to discover object boundary. We run the standard K-Means clustering algorithm with our semantic-aware initialization on the feature map from the second-to-last layer of the $\text{Enc}_I(\cdot)$ to obtain $K$ clustering results in $\mathbf{C}_{1:K}$. Each cluster $\mathbf{C}_k \in \mathbf{C}_{1:K}$ is represented as $\{0, 1\}^{hw}$, where $\mathbf{C}_k^{ij} = 1$ means that the point $(i, j)$ belongs to the $k$-th cluster.

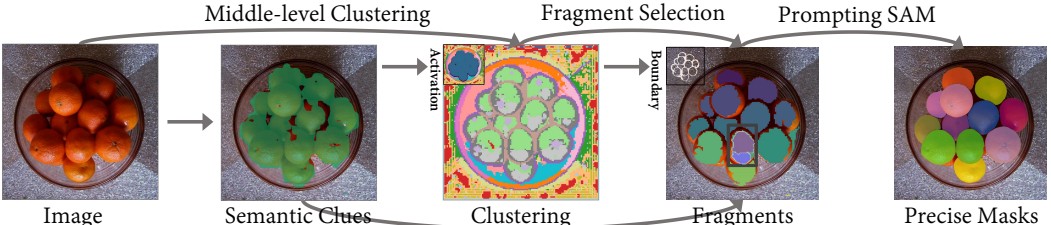

Figure 3: **Zip: Multi-object Discovery with No Supervision.** Zip follows a classification-first-then-discovery approach, consisting of four steps: 1) Classification first to obtain semantic clues provided by CLIP, where the semantic clues indicate the approximate activation regions of potential objects. 2) Clustering on CLIP's features at a specific intermediate layer to discover object boundaries with the aid of our semantic-aware initialization. The semantic-aware initialization leverages semantic activation to automatically initialize clustering centers and determine the number of clusters. 3) Localization of individual objects by regrouping dispersed clustered fragments that have the same semantics, all while adhering to the detected boundaries. 4) Prompting SAM for precise masks for each individual object.

- Localization of individual objects via fragment selection. To find each individual object, we further identify the clusters representing boundaries within the clustering results $\mathbf{C}_{1:K}$ and regroup the scattered clustering fragments for each object. Specifically, we define the boundary score metric and utilize the objective function Equ. 3 to detect boundaries and facilitate the grouping, which reformulates the instance segmentation task into a fragment selection problem and employ the activation map $S_T$ to help approximate the selection in an annotation-free manner (Sec. 3.4).

- Prompting SAM for precise masks. Although preliminary localization for individual objects is extracted, the segmentation masks generated by clustering are very coarse. Therefore, we prompt SAM to attain finer segmentation results by initially prompting it with the preliminary bounding boxes and subsequently with the center points of the refined boxes. See Appendix E for the analysis of the prompting method.

## 3.4 LOCALIZATION OF INDIVIDUAL OBJECTS VIA FRAGMENT SELECTION

In this section, we localize objects from the clustering results by framing object detection and instance segmentation as a fragment selection problem.

**Motivation and Observation**: 1) Clustering results $\mathbf{C}_{1:K}$ can be transformed into superpixel-like fragments by considering the connectivity to eight neighbors in the image of points in each cluster, as shown in "Fragments" in Figure 3. 2) A specific set of fragments can correspond to a specific object. In Figure 3, the two fragments in the bounding box coalesce to form the instance "orange". Thus, we transform the instance segmentation task into a fragment selection task. For symbolic consistency, without loss of generality, we assume that each cluster $\mathbf{C}_k$ is a connected fragment. If this condition is not met, we treat each connected fragment within the cluster as a new cluster.

**Fragment Selection.** Specifically, we select the fragment according to the activation map $S_T \in \mathbb{R}^{H \times W}$ and the clustering results $\mathbf{C}_{1:K}$, where the former roughly activates scattered fragments for objects of interest, while the latter is used to group fragments into each individual instance. ***First***, we identify fragments that belong to the category $\mathcal{T}$ but are not on the boundary regions. The objective function is:

$$\mathcal{L}_{\mathrm{FS}}(X^{\mathrm{T}}) = \sum_{k=1}^{K} \overbrace{\langle \mathbf{C}_k S_{\mathrm{T}}, \mathbb{1}(X_k^{\mathrm{T}}) \rangle}^{\text{cluster-region activation score}} - \langle \mathbf{C}_k, \mathbb{1}(X_k^{\mathrm{T}}) \rangle - \underbrace{\mathbf{f}(\mathbf{C}_k)/\langle \mathbf{C}_k, \mathbb{1}(X_k^{\mathrm{T}}) \rangle}_{\text{boundary score metric}} \quad (3)$$

where the $X_k^{\mathrm{T}} \in \mathbb{R}^{H \times W}$ is the indicator variable where $X_k^{\mathrm{T}} = 1$ signifies that the $k$-th cluster is identified to belong to category $\mathcal{T}$ and is not on the object boundary, $\langle \cdot, \cdot \rangle$ represents the inner product of matrices, and the $\mathbf{f}(\mathbf{C}_k)$ returns the area size of $\mathbf{C}_k$'s smallest enclosing bounding box. The term cluster-region activation score encourages clusters belonging to the category $\mathcal{T}$ to be selected. The term boundary score metric penalizes clusters on the object boundaries. It is computed as the

| Method | Annotation Free | AR 10 | AR 100 | $AP_s$ | $AP_m$ | $AP_l$ | AP 75 | AP 50 | AP |
|---|---|---|---|---|---|---|---|---|---|
| *COCO class-agnostic mask* | | | | | | | | | |
| 1 SAM-B (Kirillov et al., 2023) | ✓ | 10.1 | 36.9 | 1.4 | 3.3 | 5.9 | 2.7 | 4.6 | 2.6 |
| 2 SAM-H (Kirillov et al., 2023) | ✓ | 17.4 | 44.5 | 0.9 | 6.1 | 15.0 | 6.9 | 10.4 | 6.6 |
| 3 DINOv2[†] (Oquab et al., 2023) | ✓ | 2.0 | – | 0.8 | 2.2 | 1.9 | 0.6 | 1.7 | 0.8 |
| 4 CutLER[‡] (Wang et al., 2023) | ✓ | 16.5 | – | 2.4 | 8.8 | 24.3 | 9.2 | 18.9 | 9.7 |
| 5 Ours (SAM-B) | ✓ | 15.8 | 29.8 | 2.8 | 16.3 | 21.7 | 12.0 | 19.0 **+14.4** | 11.3 **+8.7** |
| 6 Ours[‡] (SAM-B) | ✓ | 23.1 | 35.1 | 5.9 | 19.0 | 31.3 | 15.3 | 24.3 **+19.7** | 15.1 **+12.5** |
| *COCO class-aware mask* | | | | | | | | | |
| 1 CLIP+SAM-B (Li et al., 2023) | ✓ | 17.9 | 23.8 | 0.9 | 5.4 | 10.0 | 2.8 | 5.7 | 3.0 |
| 2 CLIP+SAM-H (Li et al., 2023) | ✓ | 21.9 | 27.6 | 1.2 | 7.2 | 13.2 | 4.1 | 7.5 | 4.2 |
| 3 DINOv2[†] (Oquab et al., 2023) | ✓ | 2.8 | – | 0.2 | 2.7 | 2.3 | 1.0 | 2.3 | 1.2 |
| 4 CutLER[‡] (Wang et al., 2023) | ✓ | - | - | 2.7 | 7.4 | 17.1 | 8.2 | 15.7 | 8.5 |
| 5 Ours (SAM-B) | ✓ | 21.9 | 31.2 | 4.8 | 13.1 | 20.1 | 12.0 | 20.0 **+14.3** | 11.8 **+8.8** |
| 6 Ours[‡] (SAM-B) | ✓ | 30.3 | 39.2 | 5.0 | 19.6 | 31.7 | 19.1 | 30.1 **+24.4** | 18.2 **+15.2** |

Table 1: **Annotation-free instance segmentation** on the COCO val2017. For a fair comparison, we keep the same settings as the previously best-performing CutLER in the self-training setting, and Zip outperforms all previous methods on all evaluation metrics. [‡]: with self-training, [†]: our re-implemented based on Ncut (Shi & Malik, 2000).

ratio between the area of the bounding rectangle and the area of the object itself because it indicates the potential of becoming a boundary. In our implementation, we found that the heuristic solution for the Equ. 3 works well as follows:

$$
X_k^{\mathrm{T}} = \begin{cases} 1, & \text{if } \frac{\langle \mathbf{C}_k S_{\mathrm{T}}, \mathbb{1} \rangle}{\langle \mathbf{C}_k, \mathbb{1} \rangle} \geq \theta_1 \quad \text{and} \quad \frac{\mathbf{f}(\mathbf{C}_k)}{\langle \mathbf{C}_k, \mathbb{1} \rangle} \leq \theta_2 \\ 0, & \text{otherwise} \end{cases}
\tag{4}
$$

where we decouple the two terms in Equ. 3, and $\theta_1$ and $\theta_2$ are hyper-parameter set to $0.3$ and $3$ for all the experiments. ***Second***, we execute a standard connected graph algorithm (Caron et al., 2021) on the identified clusters $\sum_{k=1}^{K} \mathbf{C}_k \mathbb{1}(X_k^T)$ to obtain the instance segmentation results. We naturally classify these instances to category $\mathcal{T}$ with the confidence score being the average activation score within the regions of instances.

## 4 EXPERIMENTS

We evaluate our approach on several object detection and instance segmentation settings, including annotation-free object localization, detection, and segmentation in Section 4.1 and open-vocabulary object detection in Section 4.2. In Section 4.3, we show that our Zip can be an effective annotation-free initialization for supervised object detection and segmentation tasks. Finally, we conduct the analysis and ablation experiments in Section 4.4.

**Implementation Details.** Zip does not utilize any instance annotations and in-domain image annotations aside from fine-tuning downstream tasks in Section 4.3. In order to match the model size utilized by previous methods (Siméoni et al., 2021; Wang et al., 2023; Van Gansbeke et al., 2022), we choose the RN50x64 CLIP model (Radford et al., 2021), adjusting the image size to $2048 \times 2048$ to achieve superior clustering results. For a fair comparison, we use the same prompting method to employ SAM-B to refine original results of all compared methods.

**Self-training Details.** Our method is compatible with any architecture of object detectors, such as DETR (Carion et al., 2020) or Mask R-CNN (He et al., 2017). For a fair comparison with previous methods (Wang et al., 2023), we adopt the same detection architecture of CutLER (Wang et al., 2023). Unless specifically noted, we employ the Cascade Mask R-CNN by default in all experiments. We maintain the identical training strategy as CutLER. Please refer to Appenidx A for more implementation and self-training details.

### 4.1 Annotation-free Object Localization, Detection, and Segmentation

We conduct the annotation-free experiment on COCO (Lin et al., 2014) shown in Table 1, and an additional experiment on Pascal VOC (Everingham et al., 2010) in Appendix D. We employ the same dataset partitioning on COCO dataset as previous methods and evaluate the standard COCO metrics. The primary methods we compare include: **1)** the various variants of SAM and the naive combination of SAM and CLIP; **2)** The previous state-of-the-art unsupervised object detection technique utilizing the DINO, *i.e.*, CutLER. Additionally, we list fully-supervised baselines for reference. Notably, for unsupervised methods originally not requiring SAM, to ensure a fair comparison, we also refine their outputs by SAM. Regarding the methodologies related to SAM, we introduce how we tailor SAM for parallel computation, supplemented with pseudocode in Appendix A.1.

**Comparison on class-agnostic setting**: **1)** SAM-B demonstrates superior object localization capabilities compared to DINO (+36.9%) on AR100 by pretraining on a large amount of segmentation masks. Furthermore, SAM-H with a larger model size further improves the AR100 (+7.6%). **2)** However, both SAM-B and SAM-H perform subpar in terms of the Average Precision (AP), showing a significant gap between the edge-oriented segmentation and the downstream instance segmentation. We conduct additional analysis and illustrate why SAM fails to yield satisfactory results on the COCO dataset in Appendix B. **3)** Compared with SAM, our method without self-training surpasses SAM by +8.7% on AP, and after self-training, the lead extends to +12.5%. This improvement demonstrates that our Zip effectively applies the foundation models to the downstream instance segmentation task in a zero-shot manner. **4)** Compared with previously best-performing CutLER (Wang et al., 2023), which leverage DINO's ability to discover object. Our Zip, even devoid of self-training, achieves higher AP (+1.6%) than CutLER subjected to self-training. Employing the same self-training, we surpass the state-of-the-art unsupervised method by a margin of +5.4% AP.

**Comparison on class-aware setting**. To ensure a fair comparison, we employ the identical CLIP model to classify the segmentation masks extracted by all compared methods. The even more significant improvement compared to the class-agnostic comparison shows the effectiveness of our classification-first-then-discovery pipeline. Specifically, we surpass CutLER by a margin of +14.4% on AP50 and +9.7% on AP. In comparison, the previous CLIP+SAM (Li et al., 2023) results in a less-than-ideal performance, highlighting the non-trivial nature of utilizing CLIP and SAM for annotation-free instance segmentation.

Furthermore, our Zip model consistently and significantly improves annotation-free object detection on the Pascal VOC dataset (Table 8 in Appendix D). It enhances the state-of-the-art AP50 by +10.2% in the class-agnostic setting and +15.6% in the class-aware setting. In addition, we compare Zip with semantic segmentation models (with text-image pairs for pretraining) combined with SAM, achieving significant performance improvement on annotation-free object detection, as detailed in Appendix I.

### 4.2 Open-vocabulary Object Detection

Open-vocabulary object detection (OVD) methods are trained using annotations of objects from base categories but are required to detect and recognize objects belonging to both base and novel categories. Without resorting to a single COCO image in either base or novel categories, our annotation-free and training-free Zip achieves comparable performance to OVD methods trained on annotations of base categories. For instance, our Zip achieves comparable performance to ViLD (Gu et al., 2021) in a training-free and zero-shot manner. Following self-training, it not only significantly outstrips the baselines (Zhong et al., 2021; Zhou et al., 2022d; Shi & Yang, 2023a) (+12.5%) but also approaches a performance close to the current state-of-the-art models CORA+ (Wu et al., 2023). In addition, we also observe a performance drop from the novel category (27.1%) to all COCO 80 categories (20.0%), which is analyzed in Appendix G.

| Method | AP50$_{novel}$ |
|---|---|
| ***with annotation*** | |
| ViLD (RN50) | 27.6 |
| RegionCLIP (RN50) | 26.8 |
| Detic (RN50) | 27.8 |
| EdaDet (RN50) | 37.8 |
| CORA+ (RN50X4) | 43.1 |
| ***w/o any annotation*** | |
| **Ours** (RN50) | **23.5** |
| **Ours** (RN50x64) | **27.1** |
| **Ours**[‡] (RN50) | **40.3** |

Table 2: **OVD results on COCO.** We experiment with transductive settings where all category names are known.

### 4.3 Few-shot and Label-efficient Tuning

We now evaluate Zip as a pretraining model for training object detection and instance segmentation models following the CutLER setting. We use the self-supervised Zip to initialize the detector and fine-tune it on varying amounts of labeled data on the COCO dataset. The comparisons between Zip with the unsupervised object detection method CutLER and the self-supervised MoCo-v2 (Chen et al., 2020) are shown in Figure 4. We report both box AP and mask AP of models using from 1% to 30% labeled data. Under 1% of labeled data, our method's performance stands at 21.1% box AP, while MoCo-v2 and CutLER exhibit perfor-

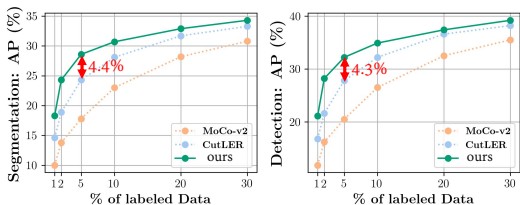

Figure 4: **Label-efficient Tuning on COCO.** We initialize the weights derived from various self-supervised methods and then fine-tune them on varying proportions of labeled data.

mances of 16.8% and 11.8%, demonstrating a +4.3% improvement of our method. With the increment in labeled data, our method continues to improve, reaching a performance of 39.2% with 30% labeled data. These findings denote that our method consistently outperforms in different volumes of labeled data. Relative to MoCo-v2 and CutLER, our method demonstrates a more significant improvement with less labeled data.

## 4.4 ANALYSIS OF OUR PIPELINE

We analyze the impact of each component on Zip and the robustness of the detector selection on the Pascal VOC val set. (1) Our baseline employs the "Segment Anything" mode from SAM (Kirillov et al., 2023) to extract proposals from the entire image, which CLIP then classifies. Although SAM is pre-trained on one billion mask data, its high AR yet low AP underscores its limitations. (2) We utilize the method from DINO (Caron et al., 2021) on CLIP semantics, yielding modest results due to the absence of instance-level information. (3) However, with the further utilization of clustering information and boundary removal to acquire instance information, the performance escalated by +20.4%. Note that we only discovery boundaries in specific layer. (4) Refining the results on this basis using SAM improved the result by +4.9%, substantiating the essentiality of each module in our methodology. (5) For any given detector architecture, our generated annotations consistently yield stable performance, with the optimal combination achieving the 52.1% AP50.

| Analysis on Our Components | AP50 | AR100 | Analysis on the Architecture | AP50 | AR100 |
|---|---|---|---|---|---|
| "SAM + CLIP" | 16.6 | 51.2 | "DETR" | 44.9 | 38.9 |
| "CLIP semantic only" | 2.2 | 3.6 | "Mask R-CNN" | 46.1 | 40.3 |
| "+ boundary detection" | 22.4 | 28.4 | "Cascade Mask R-CNN" | 50.3 | 46.1 |
| "+ two-step SAM" | 27.3 | 32.8 | "+RN101" | 52.1 | 47.6 |

Table 3: **Analysis of component in our pipeline and its robustness across different detectors.** Further analysis regarding the classification-first-then-discovery pipeline, general and stable clustering results for boundary discovery, and algorithm complexity are given in the Appendix F, A, and C.

## 5 CONCLUSION

This paper introduces Zip, an annotation-free, complex-scene-capable object detector and instance segmenter. Zip combines two different foundation models CLIP and SAM to achieve a robust instance segmentation performance on downstream datasets. First, we highlight our observation that the clustering outcome from the intermediate layer output in CLIP is keenly attuned to the boundaries between objects. Building on this, we introduce our classification-first-then-discovery pipeline and convert the instance segmentation task into a fragment selection task. Our approach significantly surpasses previous methods across a wide range of settings. **Ethics Statement:** It is possible that any existing biases, stereotypes, and controversies inherent within the image-text pairs utilized for training the CLIP could inadvertently permeate our models.

**Acknowledgment:** This work was supported by the National Natural Science Foundation of China (No.62206174) and MoE Key Laboratory of Intelligent Perception and Human-Machine Collaboration (ShanghaiTech University).

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

## A  APPENDIX

In Appendix, we provide additional information regarding,

- Implementation Details (Appendix A)
- Why does SAM Fail in AP? (Appendix B)
- Clustering with Semantic-aware initialization (Appendix C)
- Additional Results on Pascal VOC (Appendix D)
- Is SAM a Free Lunch? (Appendix E)
- Why Classification then Discovery (Appendix F)
- A Closer Look at COCO 80 Categories (Appendix G)
- Qualitative Results (Appendix H)
- Compared with Weakly supervised Semantic Segmentation methods (Appendix I)
- Limitations (Appendix J)
- Datasets License Details (Appendix K)
- Understanding the Process of Clustering and Fragment Selection (Appendix L)
- Comparison between DINO and CLIP feature (Appendix N)

## A  IMPLEMENTATION DETAILS

### A.1  SAM AUTOMATICALLY MASK GENERATE IN PARALLEL

The official implementation of SAM's SamAutomaticMaskGenerator does not support the parallel processing of multiple images. Utilizing SamAutomaticMaskGenerator to extract annotations from a large-scale dataset proves to be time-consuming. For instance, extracting proposals from the entire COCO dataset requires approximately 48 hours. We dissected the various modules within SamAutomaticMaskGenerator, modified a portion of the code, and by uniformly scattering points within the images, enabled the model to accept parallel inputs, markedly accelerating the process. In comparison to the prior implementation, our version merely requires 15 minutes on 8 A100 GPUs to complete the task.

```python
# SAM is comprised of three components: an image encoder, a prompt
    encoder, and a mask decoder
# We take batch of images as inputs
Batch_inputs = [...]

# Encode the image:
image_features = SAM.image_encoder(Batch_inputs)

# Generate grid point prompt:
num_intervals = 32

# Create a grid
x = torch.linspace(0, 1, num_intervals)
y = torch.linspace(0, 1, num_intervals)
x_grid, y_grid = torch.meshgrid(x, y)
points = torch.cat((x_grid.reshape(-1, 1), y_grid.reshape(-1, 1)), dim
    =1).unsqueeze(1)
labels_torch = torch.ones_like(points[:, :, 0])

prompt_input = (points, labels_torch)
sparse_embeddings, dense_embeddings = SAM.prompt_encoder(
    points=prompt_input,
    boxes=None,
    masks=None,
)
```

```
24  # Generate Mask in parallel
25  masks, scores, logits = SAM.mask_decoder(
26       image_embeddings=image_features,
27       image_pe=SAM.prompt_encoder.get_dense_pe(),
28       sparse_prompt_embeddings=sparse_embeddings,
29       dense_prompt_embeddings=dense_embeddings,
30       multimask_output=False,
31  )
```

Modifications on the mask decoder.

```
1  # Borrow from SAM GitHub issue and MedSAM
2  # Line 126 Before
3  src = torch.repeat_interleave(image_embeddings, tokens.shape[0], dim=0)
4  # Line 126 After
5  if image_embeddings.shape[0] != tokens.shape[0]:
6      src = torch.repeat_interleave(image_embeddings, tokens.shape[0], dim
       =0)
7  else:
8      src = image_embeddings
```

We also compared the AR10 and AR100 among different methods in Table 4. We discovered that the complex yet non-parallelizable post-processing indeed augmented the Average Recall (AR), however, our own model also surpassed the original SAM in terms of AR10.

| Method | $AR_{10}$ | $AR_{100}$ |
|---|---|---|
| SAM-B E32 | 11.4 | 39.6 |
| SAM-H E32 | 18.5 | 49.5 |
| SAM-B-Parallel E32 | 10.1 | 36.9 |
| SAM-H-Parallel E32 | 17.4 | 44.5 |
| **Ours** | 15.8 | 29.8 |
| **Ours**[‡] | 23.1 | 35.1 |

Table 4: Comparison of different SAM models at varying sampling points, contrasting the original and parallelized versions in terms of AR (Average Recall).

## A.2  SAM+CLIP IN PARALLEL

Inspired by F-VLM(Kuo et al., 2022), in order to avoid redundant cropping of images and input the entire proposal bounding box region into CLIP, we use RoIAlign to feed the corresponding visual features into the pooling layer of CLIP. We also compared the SAM+CLIP in parallel with the naive method which crops the proposal from the image and resizes it to 224 then feeds it into CLIP. SAM+CLIP in Parallel boosted efficiency by a factor of seven without compromising performance.

```
1  # load CLIP
2  clip_model, preprocess = load("RN50")
3  clip_model_ = list(clip_model.visual.children())
4  backbone = nn.Sequential(*(clip_model_[:-1]))
5  pooling = nn.Sequential(clip_model_[-1])
6
7  # After obtaining several proposals
8  proposal_bbox = [...] # xyxy
9
10 # After obtaining clip backbone feature
11 clipout = backbone(preprocess(image))
12
13 # ROI bbox feature feeds into pooling
14 from torchvision.ops import roi_align
```

```
15 outputsize = [7,7]
16 index = torch.zeros(proposal_bbox.shape[0]).unsqueeze(-1)
17 interest_bbox = torch.cat((index, proposal_bbox), dim = -1)
18 interest_roi_feature = roi_align(clipout, interest_bbox, outputsize,
       spatial_scale=16, aligned = True)
19
20 clip_feature = pooling(interest_roi_feature)
21 clip_feature = clip_feature / clip_feature.norm(dim=-1, keepdim=True)
```

| Method | AP | Efficiency |
|---|---|---|
| SAM+CLIP Crop | 3.2 | 1X |
| SAM+CLIP Parallel | 3.0 | 7.5X |

Table 5: Comparison of AP and Efficiency of Parallel implementation in COCO.

### A.3 SELF-TRAINING DETAILS

We provide a detailed implementation for the self-training phase. Note that we adhere to the strategies outlined in CutLER (Wang et al., 2023), with the singular exception that we employ a single round of self-training, in contrast to the three rounds utilized in CutLER. By default, we employ the Cascade Mask R-CNN (He et al., 2017) with a ResNet-50 (He et al., 2016) backbone. We initialize the weights using the self-supervised pre-trained DINO (Caron et al., 2021) model and carry out the process for 160K iterations with a batch size of 16. For data augmentation, we adopt a variant of the copy-paste augmentation identical to CutLER, which entails randomly downsampling the mask with a scalar uniformly sampled between $0.3$ and $1.0$. For the optimizer, we leverage SGD with an initial learning rate of $0.005$ which is diminished by a factor of $5$ after $80K$ iterations, a weight decay of $5x10^{-5}$, and a momentum of $0.9$.

### A.4 LABEL-EFFICIENT TUNING DETAILS

We start from a weight initialization obtained through self-supervised training. To ensure a fair comparison with CutLER, we adopt the same training strategies of CutLER, which adjusts the learning rates according to varying quantities of tuning data, as illustrated in Table 6.

| % | Max Iters | Diminishing Steps | WarmUp Iters |
|---|---|---|---|
| 1 | 3,600 | 2,400 / 3,200 | 1,000 |
| 2 | 3,600 | 2,400 / 3,200 | 1,000 |
| 5 | 4,500 | 3,000 / 4,000 | 1,000 |
| 10 | 9,000 | 6,000 / 8,000 | 0 |
| 20 | 18,000 | 12,000 / 16,000 | 0 |
| 30 | 27,000 | 18,000 / 24,000 | 0 |

Table 6: Summary of training strategies of CutLER and ours.

### A.5 DATASETS DETAILS

We have employed two commonly used datasets in object detection and instance segmentation, COCO (Lin et al., 2014) and Pascal VOC (Everingham et al., 2010), as benchmarks. To compare with previous methods, we use the same test sets as previous methods (Wang et al., 2022b; 2023; Siméoni et al., 2021; Van Gansbeke et al., 2022). In Table 7, we enumerate the specific conditions of the training and test sets used under different settings. Configuration A and B are applied in Table 1, while Configuration C is utilized in Section 4.4.

| settings | training data | testing data | #images | segmentation |
|----------|--------------|--------------|---------|--------------|
| A | COCO `train2017` split | `val2017` split | 5,000 | ✔ |
| B | COCO `train2017` split | `trainval07` split | 9,963 | ✘ |
| C | VOC `train07` split | `val07` split | 2,510 | ✘ |

Table 7: Datasets details under different settings.

## B WHY DOES SAM FAIL IN AP ?

We visualized the ground truth alongside the top 10/30 predictions based on SAM's confidence scores. We discerned that SAM prefers to predict smooth, regular pixel blocks, or smaller objects since SAM's edge-oriented property. These objects, in the prevailing dataset annotations, are highly likely not to be the targeted objects for detection, hence SAM yielded a comparatively low Average Precision (AP).

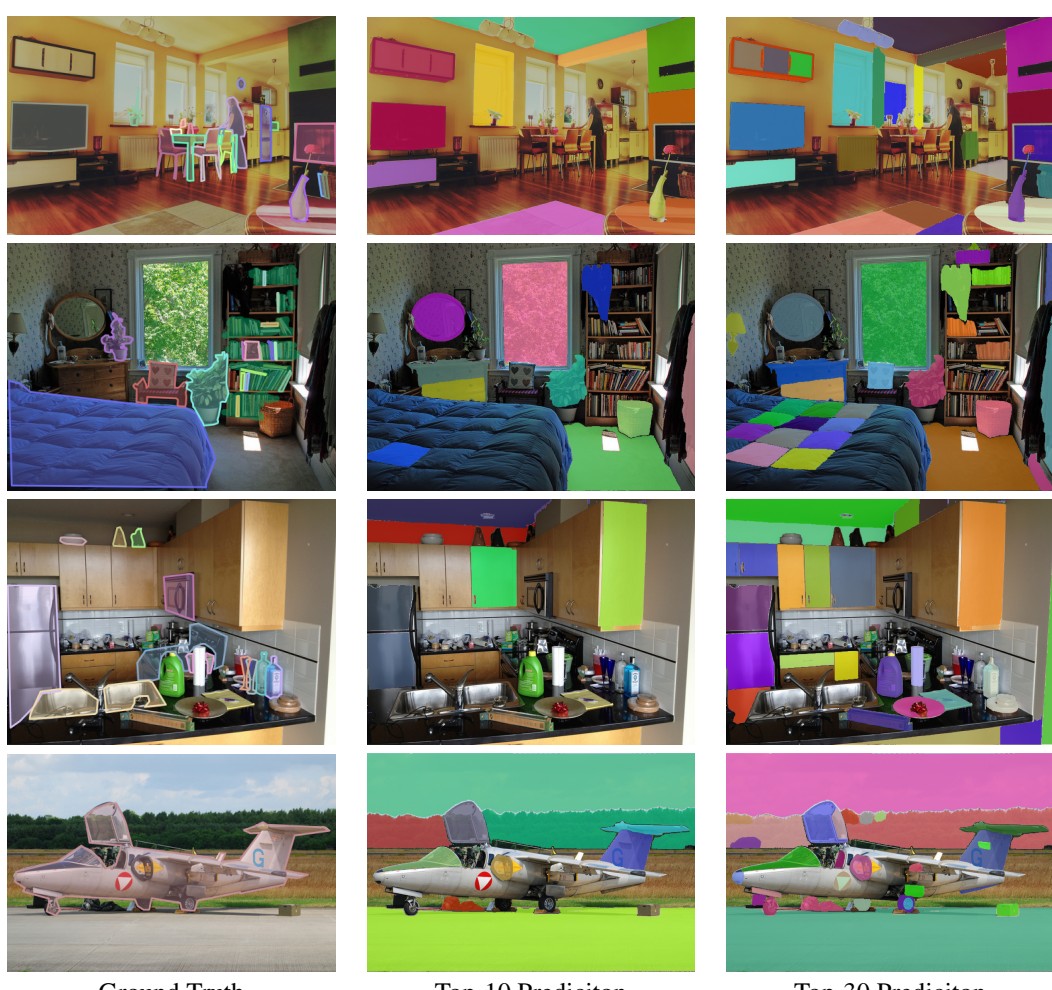

Ground Truth            Top-10 Prediciton            Top-30 Prediciton

## C CLUSTERING WITH SEMANTIC-AWARE INITIALIZATION

**Motivation and Observation.** Although the activation map $S_T$ offers semantic information at the dense level, two serious drawbacks hinder it from performing accurate instance segmentation: (1) **Rough semantic segmentation results.** Since mask-level annotations are never exposed to CLIP for training, the activation map of CLIP fails to accurately activate the local regions especially object boundaries, resulting in coarse localization. (2) **Lack of instance information.** As the oranges are

shown in Figure 3, due to their close proximity, all instances are in a single connected semantic segment and cannot be distinguished. To overcome these shortcomings, our key observation is that after clustering the features of a specific middle layer in CLIP, not only can pixels be grouped into "super-pixels/fragments", but what is even more astonishing is that it is highly sensitive to object edges, resulting in the formation of object boundaries as shown in the Figure 3(Clustering).

**Semantic-Aware Initialization.** To obtain impressive clustering results via K-Means (Hartigan & Wong, 1979), we utilize the activation map to initialize the clustering centers and determine the number of clusters as the activation map serves a strong prior where the objects localize. Our principle is that active regions are prioritized over those that are far away from them, meanwhile, the cluster centers should be more located near the edges between the active regions and the inactive ones to identify the boundary. To implement the principle, we first convert the activation map into a binary map $M$ by applying a threshold $\tau$:

$$M^{ij} = \begin{cases} 1, & \text{if } S_{\text{T}}^{ij} \geq \tau \times \text{mean}\,(S_{\text{T}}) \\ 0, & \text{otherwise} \end{cases} \tag{5}$$

where $\text{mean}(\cdot)$ represents the computation of the average activation score for the activation map, $\tau$ is set as $0.7$ for all datasets. Then, we obtain the smallest enclosing bounding box of the region where $M^{ij} = 1$ and specified to its top left and bottom right coordinates as $(\mathbf{x}_{\min}, \mathbf{y}_{\min})$ and $(\mathbf{x}_{\max}, \mathbf{y}_{\max})$, respectively. The bounding box divides the active regions and the inactive ones.

To align with our principle that the cluster centers are expected to be near the edges between the active regions and the inactive ones, we compute the distance $d^{ij}$ from each point $(i, j)$ in the map to the bounding box and apply a Gaussian kernel on the distance $d^{ij}$ to obtain the heatmap $Y$:

$$d^{ij} = \min\left((i - \mathbf{x}_{\min})^2, (i - \mathbf{x}_{\max})^2\right) + \min\left((j - \mathbf{y}_{\min})^2, (j - \mathbf{y}_{\max})^2\right),$$
$$Y^{ij} = \exp\left(-\frac{d^{ij}}{2\sigma^2}\right), \tag{6}$$

where $\sigma$ is adaptive based on the size of the feature map. The score $Y^{ij}$ of the heatmap roughly reveals how close a point $(i, j)$ is to the contours of active and inactive regions. Therefore, we sample the cluster centers based on the normalized heatmap.

Regarding the number of clustering centers $K$ for one image $\mathcal{I}$, we expect it to be a positive correlation to the number of categories of interest appearing in the image because one cluster is expected to be corresponding to some regions of one category. Therefore, we empirically define it as $K = \max\left(20, (Q(\mathcal{I}) + 1)^3\right)$, where $Q(\mathcal{I})$ is:

$$Q(\mathcal{I}) = \sum_{\mathcal{T} \in \mathcal{T}_{\text{class}}} \mathbb{1}\left(\mathcal{S}_{\text{CLIP}}^{\text{img}}(\mathcal{I}, \mathcal{T}) > 0.15\right), \tag{7}$$

where $\mathcal{T}_{\text{class}}$ is the set of all interest categories.

**Clustering.** Notice that we first obtain the heatmap individually for each category $\mathcal{T}$ and then cluster once considering all categories $\mathcal{T}_{\text{class}}$. The clustering centers are sampled based on the probability normalized from the weighted sum on heatmaps for all categories. With our semantic-aware initialization, we run the standard K-Means clustering algorithm on the feature map from the second-to-last layer of the $\text{Enc}_I(\cdot)$ to obtain $K$ clustering results in $\mathbf{C}_{1:K}$. Each cluster $\mathbf{C}_k \in \mathbf{C}_{1:K}$ is represented as $\{0, 1\}^{hw}$, where $\mathbf{C}_k^{ij} = 1$ means that the point $(i, j)$ belongs to the $k$-th cluster.

**Analysis of Computational Complexity.** Our method is more computationally efficient than previous algorithms that require computing the eigenvectors of the Laplace matrix. Our main computation is the K-means clustering, whose time complexity is $K$ times the square of the number of patches. In practical implementation, compared to the previous best-performing CutLER under default settings, our algorithm generally proffers a $10\times$ increase in processing speed.

C.1 CLUSTERING RESULTS ANALYSIS.

**Clustering Results Analysis.** Figure 5 shows the impact of using CLIP's different layers and K-Means's initialization on clustering results. From left to right, the clustering features are derived from shallower to deeper layers, with the middle three stemming from the intermediate layer but with

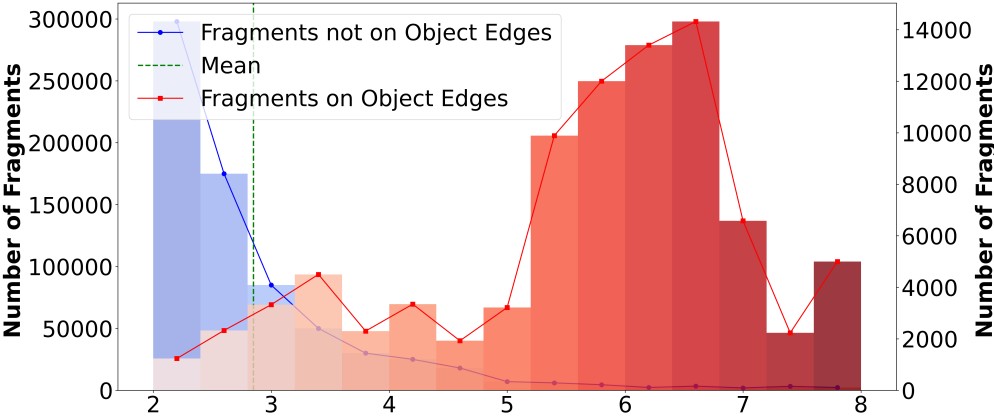

Figure 5: The influence of different layers of CLIP and initialization methods for clustering results.

different initialization methods. Layers either too shallow or too deep do not attend to information well-suited for instance segmentation, and the intermediate layer's results are not quite stable in the absence of proper initialization. Our method delivers stable clustering and boundary discovery. Since we only find CLIP's ResNet variants can clustering boundaries. we further discuss why CLIP's intermediate features have the potential for instance-level localization below.

- **Hierarchical representation structure of CNN Architecture:** The CNN representation structure with progressively increasing receptive fields exhibits a pattern where lower layers focus on local and contour details, while higher layers attend to global and semantic information. As discussed in [1], ViTs have more highly similar representations across all layers and incorporate more global information at lower layers due to self-attention. As a result, ViT's intermediate layers lose instance-level localization information.

- **Distinctive pretraining approach and training data:** The ImageNet supervised pretrained model (ResNet) is trained on object-center images for single-label classification without the requirement for learning instance-level information on complex scenes. In contrast, CLIP's pretraining approach utilizes image-text pairs, where the text could describe multi-instances present in the image and their relations. Image features must capture instance-level information in complex scenes to match highly with the text.

To prove the boundary clustering results are general in the entire COCO dataset, we further provide qualitative results across the entire dataset. We count the boundary scores of fragments that are and are not on the object edges on the COCO dataset (see Figure 13), indicating that fragments on object edges indeed have higher boundary scores in general.

Figure 6: Boundary scores of fragments on and off object edges sampled from the COCO dataset. The x-axis corresponds to the boundary score.

## D ADDITIONAL RESULTS ON PASCAL VOC

In Table 8, we give the annotation-free object detection results on the VOC dataset. For VOC, given that VOC's training set is also employed for testing, we directly utilize the overlapping 18 categories from COCO for training while maintaining tests on 20 classes. Consequently, this is not an "apple-to-apple" comparison. But considering the non-trivial improvement of $19.7\%$ in AP, it's evident that our approach boasts robust dataset transferability capabilities.

| Class-aware | | VOC Box | | |
| Model | Self-Train | AP | AP50 | AP75 |
|---|---|---|---|---|
| DINOv2[†] (Oquab et al., 2023) | | 0.9 | 2.8 | 0.4 |
| TokenCut (Wang et al., 2022b) | | 4.7 | 8.8 | 4.3 |
| CutLER (Wang et al., 2023) | | 5.4 | 9.9 | 5.8 |
| Zip (Ours) | | 16.5 | 25.5 | 17.3 |
| *vs. prev.* SOTA | | +11.1 | +15.6 | +11.5 |
| CutLER (Wang et al., 2023) | ✔ | 7.6 | 13.3 | 7.4 |
| Zip (Ours) | ✔ | 27.3 | 49.4 | 26.9 |
| Class-agnostic | | VOC Box | | |
| Model | Self-Train | AP | AP50 | AP75 |
| DINOv2[†] (Oquab et al., 2023) | | 0.7 | 2.5 | 0.3 |
| TokenCut (Wang et al., 2022b) | | 3.6 | 6.7 | 3.2 |
| CutLER (Wang et al., 2023) | | 8.5 | 10.2 | 6.6 |
| Zip (Ours) | | 14.1 | 20.4 | 14.5 |
| *vs. prev.* SOTA | | +5.6 | +10.2 | +7.9 |
| LOST (Siméoni et al., 2021) | ✔ | 6.7 | 19.8 | - |
| FreeSOLO (Wang et al., 2022a) | ✔ | 5.9 | 15.9 | 3.6 |
| CutLER (Wang et al., 2023) | ✔ | 20.2 | 36.9 | 19.2 |
| Zip (Ours) | ✔ | 20.7 | 37.7 | 20.3 |

Table 8: **Annotation-free object detection** on VOC. For a fair comparison, we keep the same settings as the previously best-performing CutLER in the self-training setting, and Zip outperforms all previous methods on all evaluation metrics.

# E   IS SAM A FREE LUNCH?

SAM (Kirillov et al., 2023), as a pre-trained foundational model for segmentation, is capable of accepting a variety of prompt inputs. One straightforward application of SAM involves its deployment to refine relatively coarse masks. We prompt SAM in two steps, in the first step, we prompt the SAM with the original predicted bounding boxes. For the second step, we prompt the SAM by the center points of boxes that output from step one. However, in our attempts, we discovered that employing SAM for refinement does not invariably yield favorable results, as shown in Figure 7. In the case illustrated in Figure 7(a), SAM aids in refining the object boundaries and replenishing any deficits in the object. However, in Figure 7(b), despite Zip providing generally accurate results, the outcomes, post SAM refinement, either overflow or become a part of the object. We hypothesize the plausible cause to be that SAM lacks the concept of object level, which could potentially lead to the phenomenon of the primary subject being lost. Despite our empirical evidence demonstrating that SAM can generally enhance segmentation results, our model's superior performance does not mainly come from SAM's segmentation capabilities. However, the question of how to utilize SAM more effectively remains open-ended.

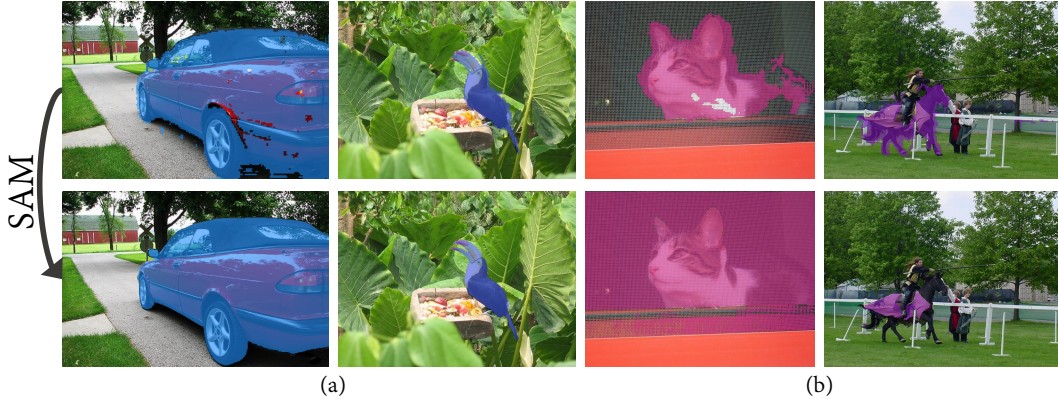

(a)                                                            (b)

Figure 7: **The impact of SAM refinement.** (a) SAM refinement has elevated the mask AP. (b) SAM refinement has diminished the mask AP.

## F WHY CLASSIFICATION THEN DISCOVERY

In our experiments, we have found that when using CLIP for zero-shot object classification, it is highly susceptible to interference from in-context information in the images. In Figure 8, we have illustrated an example called the "cat-car dilemma" phenomenon. When a cat is sitting on a car, the proposal for the car inherits the in-context information of the cat. In a supervised setting, the model might learn that because the bounding box is tighter around the car, it should be classified as a car. However, in the zero-shot scenario, we observe that the score for the "cat" category is higher. This phenomenon may be related to biases present in CLIP. How we address this phenomenon is first to perform per-pixel classification. This approach separates the semantics of the cat and the car. Subsequently, we generate masks based on semantic priors, effectively avoiding the cat-car dilemma.

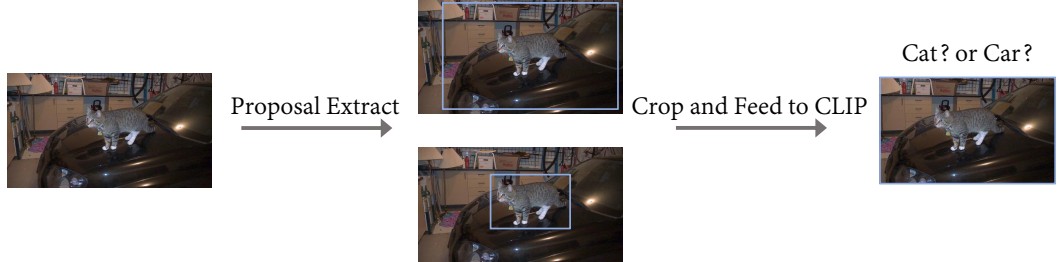

Figure 8: The cat-car dilemma.

## G A CLOSER LOOK AT COCO 80 CATEGORIES

We observe a performance drop from the novel category (27.1%) to all COCO 80 categories (20.0%). Due to the model's ability to recognize objects stemming from CLIP, in this section, we will randomly select 100 objects from each category on COCO validation set. These objects will be cropped and fed into CLIP. Similarity scores with text embeddings from each of the 80 COCO categories will be computed, resulting in the following Figure 9. The blue line represents novel objects. We observe that CLIP's recognition of categories is uneven. For certain novel categories such as "cat" and "dog," CLIP exhibits a recognition probability exceeding 90%. Surprisingly, for the common category "person", CLIP demonstrates minimal familiarity, with an accuracy rate falling below 20%. The average accuracy in the 18 novel categories is 49.8% and 43.2% for all categories, which aligns with the performance trend of our model after self-training.

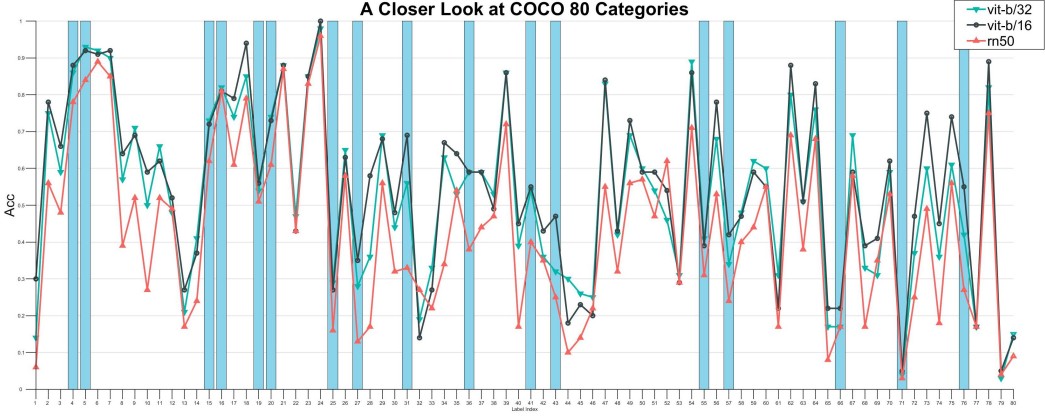

Figure 9: CLIP accuracy on 80 categories.

# H  Qualitative Results

We provide qualitative results of unsupervised Zip in Figure 10. Regardless of the complexity or ambiguity of the scene, Zip consistently performs commendably.

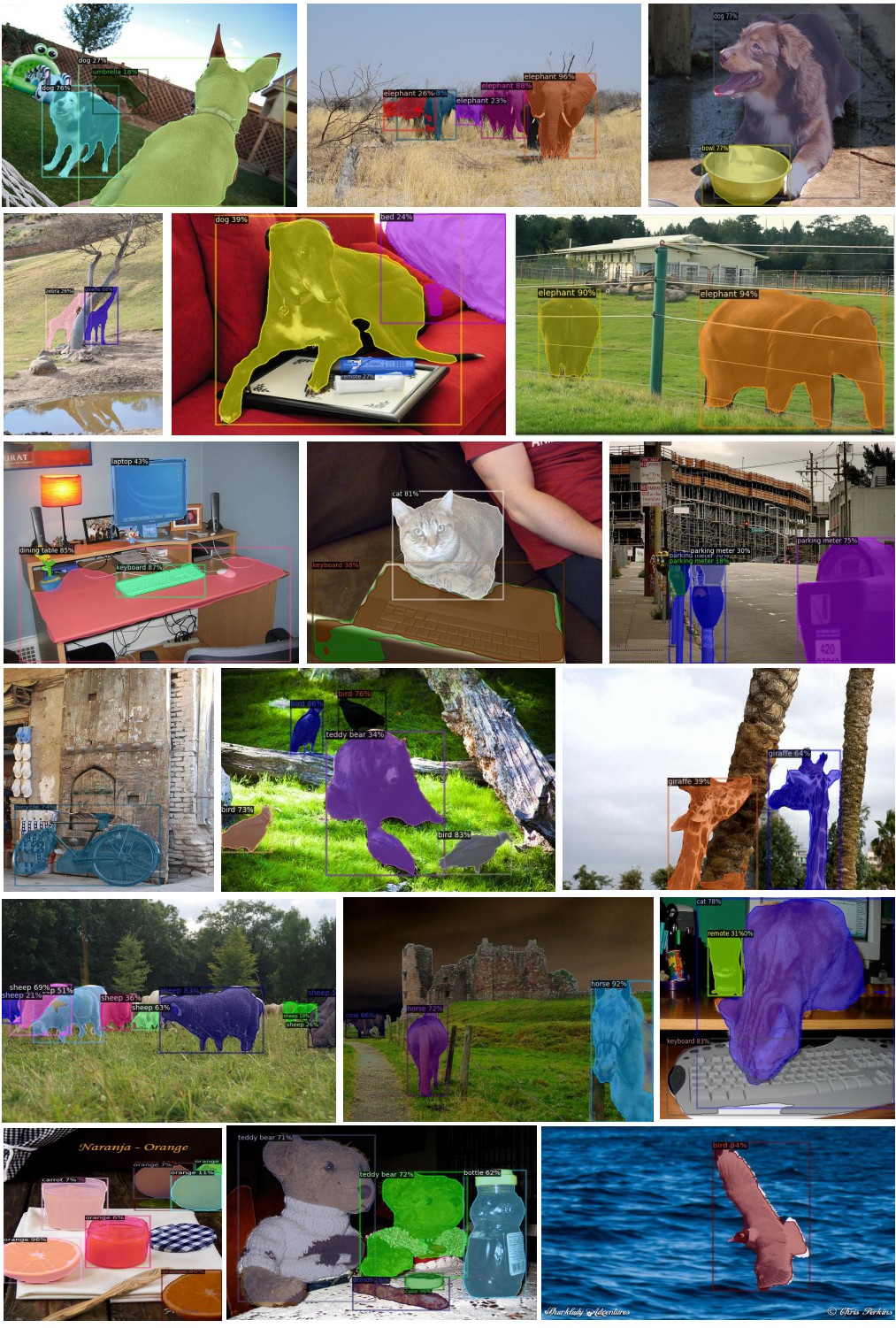

Figure 10: **Visualizations of Zip's predictions under a diverse array of scenarios.**

## I COMPARED WITH WEAKLY SUPERVISED SEMANTIC SEGMENTATION METHODS

Since CLIP was not designed for per-pixel classification, we have explored alternative weakly supervised semantic segmentation method (Xu et al., 2022; Li et al., 2023) to serve as semantic priors. However, since the clustering results from other networks do not exhibit the boundary phenomenon we have identified, the final outcome closely aligns with using CLIP alone for semantic prior. The relatively lower performance also indicates the challenge of achieving instance segmentation solely based on semantics. We also supply the outcomes achieved by substituting DINO features in place of CLIP's. The results demonstrates that the CLIP's boundary phenomenon is important for object detection.

| Semantic Method | $AP_{50}$ | $AR_{100}$ |
|---|---|---|
| GroupViT (Xu et al., 2022) + SAM-B | 5.5 | 7.2 |
| CLIP Radford et al. (2021) + SAM-B | 2.2 | 3.6 |
| CLIPSurgery (Li et al., 2023) + SAM-B | 4.7 | 6.9 |
| CLIP + DINO + SAM-B | 6.1 | 8.8 |
| **Ours** | 27.3 | 32.8 |

Table 9: Comparison of different Semantic Segmentation Method on Pascal VOC.

## J LIMITATIONS

Zip achieves twice the performance of previous methods under identical settings without employing any annotations. However, there remains a certain gap when compared with fully supervised methods. We have attempted to bridge this gap by utilizing more training data and conducting multiple rounds of self-training. However, experiments have revealed that training the model with initial objects obtained from unsupervised methods tends to trap the model as sub-optimal. The misalignment between training objectives and ground-truth annotations hinders the further improvement of self-training. Addressing this issue is a potential future work that may require the inclusion of ground-truth examples.

### J.1 ETHICAL CONSIDERATIONS AND POTENTIAL NEGATIVE SOCIAL IMPACTS

Our open-vocabulary prowess is entirely predicated upon the pre-trained foundational model, CLIP. However, given that CLIP was trained on a dataset assembled from the internet, any undesirable biases may still potentially permeate our models despite data cleansing. Any biases within the pre-trained model should be mitigated before employing our method for downstream tasks to circumvent deleterious influences. Furthermore, the detector vocabulary is flexible and can be manipulated to display racial bias during human detection. For instance, the text prompt can be set as specialized biased descriptions, which is a critical factor to be taken into account.

## K LICENSE DETAILS

All utilized codes and licenses are cataloged in Table 10. ANCSA: Attribution-NonCommercial-ShareAlike, CCA: Creative Commons Attribution.

Table 10: The used Datasets/Model Weights/Codes and License.

| Datasets/Weights/Codes | citations | License |
|---|---|---|
| COCO | Lin et al. (2014) | CCA 4.0 License |
| PASCAL VOC | Everingham et al. (2010) | NO License |
| https://github.com/openai/CLIP | Radford et al. (2021) | MIT License |
| https://github.com/facebookresearch/DINO | Caron et al. (2021) | Apache License |
| https://github.com/facebookresearch/CutLER | Wang et al. (2023) | ANCSA4.0 |

## L  UNDERSTANDING THE PROCESS OF CLUSTERING AND FRAGMENT SELECTION

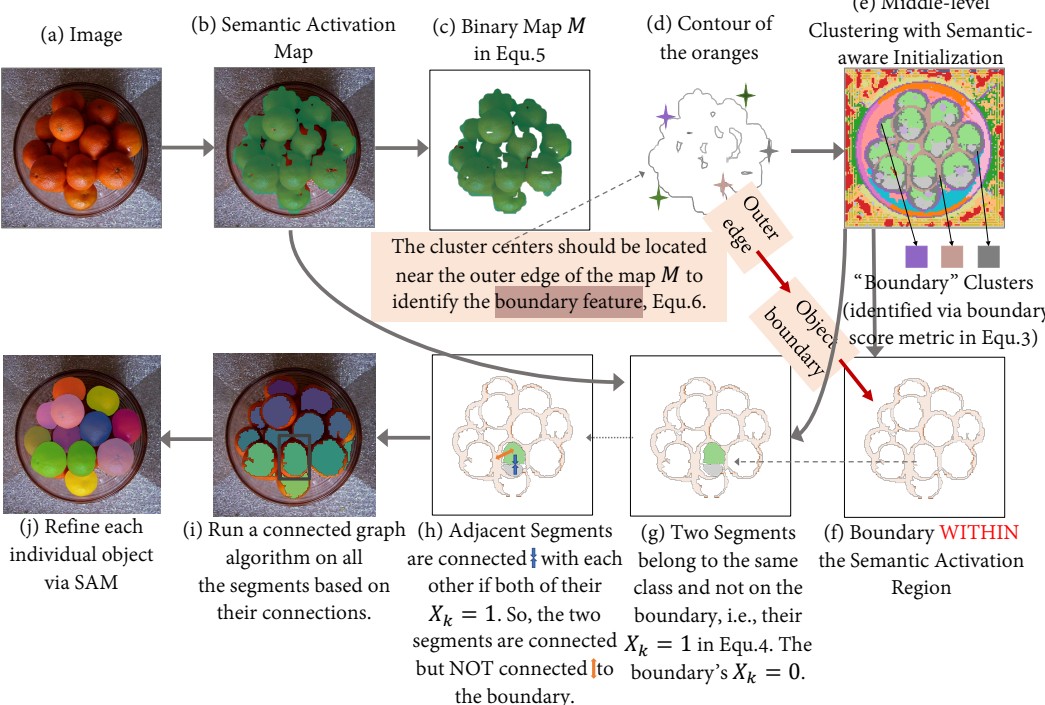

Figure 11: The visualization for an explanation of the clustering motivation and process.

**The Importance of Semantic-Aware Initialization and Its Motivation:** Semantic-aware initialization leverages the semantic activation map to **automatically initialize clustering centers and determine the number of clusters.**

- **Importance:** Without our semantic-aware initialization, K-Means fails to produce general and reliable clustering results for delineating object boundaries. As shown in Figure 5, despite clustering the same intermediate features in CLIP, it is only when utilizing the proposed initialization technique that stable clustering and boundary discovery are achieved.

- **Motivation:** By initializing certain clustering centers **near the outer edges** of the semantic activation map, boundaries between objects **within** the activation map can be clustered and outlined. This is because whether they are outer edges or inner object boundaries, they are all edges with high feature similarity.

**The explanation of clustering process**:

- We convert the activation map into a binary map $M$ using Equation 5, as shown in Figure 11(c). This conversion enables us to make a preliminary estimation of the outer edges of the activation map, as demonstrated in Figure 11(d). While the outer edges may not distinguish individual instances of oranges, they consistently delineate the boundaries between the cluster of oranges and the background. Our intention is to leverage the edge features of these outer edges to aid in identifying additional edges, such as those between oranges, as we hypothesize a significant similarity between the outer edges and the oranges' boundaries.

- Therefore, we compute the heatmap $Y^{ij}$ in Equ. 6 to roughly reveals how close a pixel $(i, j)$ is to the outer edges. Then, we sample the cluster centers based on $Y$, **so that the cluster centers can be initialized by outer edges' features**, as shown in Figure 11(d). For the number of clusters, we roughly estimate it based on estimating the number of object categories in single images through Equ. 7. In reality, our method only needs to set the number of clusters within a certain range (15 to 30) to work effectively, thanks to our reformulation instance segmentation by fragment selection.

Our method is even robust to over-segmentation caused by a slightly larger number of clusters (as long as it is not too small) because our subsequent fragment selection can regroup over-segmented fragments into an instance.

- By performing the K-Means clustering algorithm with the above initialization, the clustering results shown in Figure 11(e) are obtained, with each cluster represented by a distinct color. The "boundary" clusters can also be identified via the boundary score metric in Equ. 3.

- In summary, we accomplish the transformation from the outer edges of the activation map to the boundaries of individual objects within the activation map!

The process of fragment selection is visualized in Figure 11(f)–(j) for a more intuitive understanding of the Section 3.4.

## M    CLUSTERING CLIP FEATURES ON THE HUMAN PART SEGMENTATION

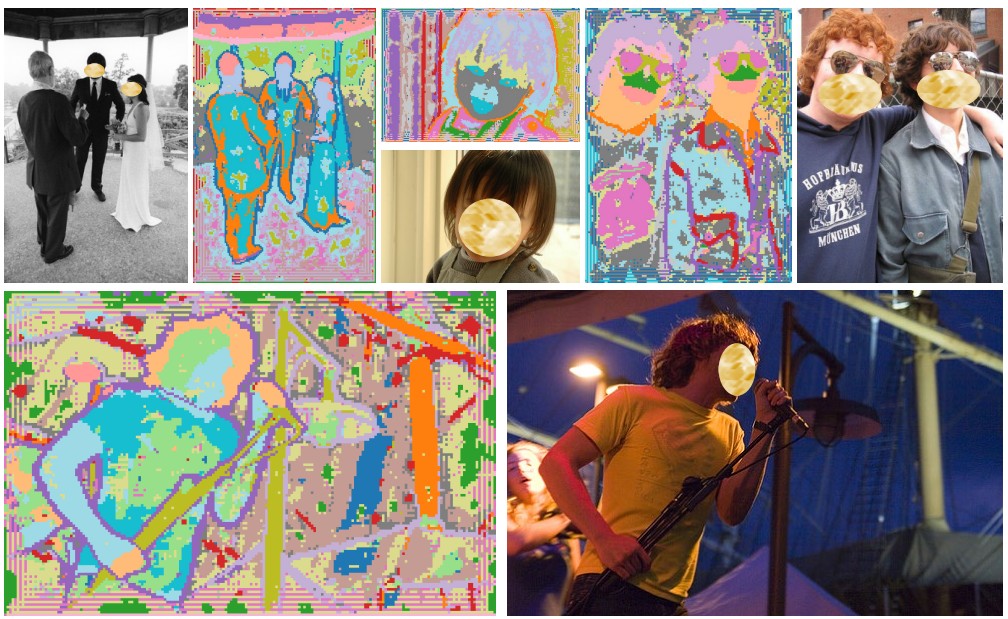

Figure 12: Clustering CLIP features on the Human Part Segmentation.

We test the clustering results by CLIP (Radford et al., 2021) on images from the Pascal VOC Part (Everingham et al., 2010) in Figure 12. The experimental results demonstrate that using our clustering method on CLIP middle-layer features yields robust clustering effects across different domains.

- The boundaries of individuals remain clear. Despite the variation in image domains, transitioning from COCO common objects to human-centric images, the boundary information of objects or individuals remains well-preserved. This highlights the robustness of CLIP features and our Zip.

- More surprisingly, the clustering results align well with human part semantics, including features like the nose, mouth, eyes, arms, and hair. This may also unveil the potential for using CLIP features to aid part segmentation in future research.

## N    COMPARISON BETWEEN DINOV2 AND CLIP FEATURE

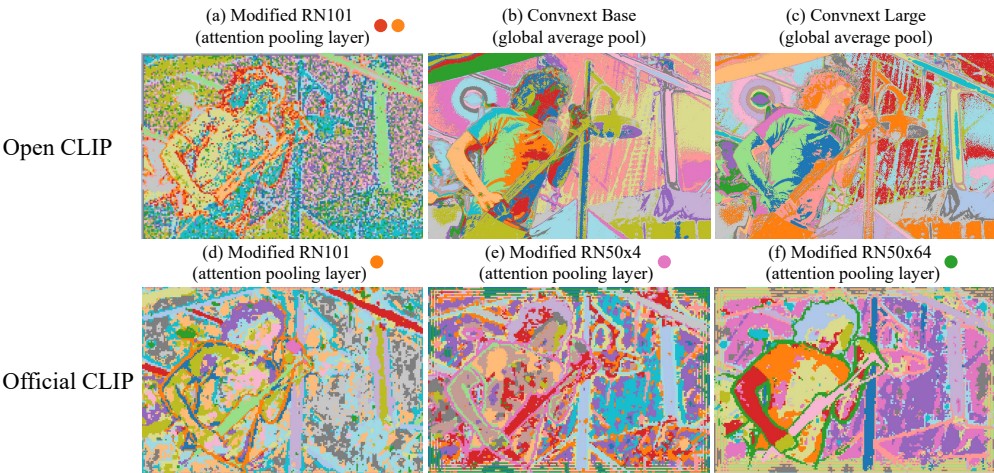

Figure 13: In the comparison of clustering the boundaries of "people" between the CLIP middle-layer feature and the DINOv2 feature, all clustering settings are kept the same.

We compare the results of the CLIP-middle layer and DINOv2 middle layer under the same clustering method. In a nutshell, DINOv2's clustering results appear smoother, with larger contiguous blocks sharing similar semantics, but it lacks instance-specific information. On the other hand, CLIP's results are comparatively "noisier" yet provide a clear delineation of object boundaries. As shown in (a)-(c), the boundaries of all individuals ("person") are clustered, with pink in (a), red in (b), and green in (c). As shown in (d)-(e), CLIP's features focus on more inconspicuous objects, such as the person in the television (d) and the three individuals in (e).

## O    COMPARISON BETWEEN OPEN CLIP AND OFFICIAL CLIP

Figure 14: In the comparison of clustering the boundaries of "people" between the official CLIP middle-layer feature and the Open CLIP feature, all clustering settings are kept the same. Boundary color also show in

We compare the results of the official CLIP-middle layer and Open CLIP middle layer under the same clustering method in Figure 14. Only model with modified attention pooling layer outline boundary. In official CLIP, attention pooling with the [CLS] token is used to obtain global image features, offering attention maps for free and allowing features with low attention scores to focus on aspects beyond "category" and semantics, such as boundaries. However, Open CLIP uses global

average pooling to acquire global image features, emphasizing that features from different spatial positions pay more attention to aspects related to "category" and semantics.

