# OpenReview forum: "The Devil is in the Object Boundary: Towards Annotation-free Instance Segmentation using Foundation Models"
_ICLR.cc/2024/Conference — ICLR 2024 poster_

### Official Review · Reviewer_CLaV · 2023-10-30

**Soundness:** 4 excellent
**Presentation:** 4 excellent
**Contribution:** 4 excellent
**Rating:** 5
**Confidence:** 4

**Summary:**

This paper proposed the annotation-free instance segmentation method using vision foundation models (i.e., SAM and CLIP). They found that SAM shows high recall rates but relatively low precision. Also, DINO is suitable to obtain salient regions but is not suitable to obtain instance-level masks. To address the challenges, they discovered that clustering the features of CLIP's specific middle layer can be effectively used for adequate prompts for SAM. The class of each instance mask is classified using the CLIP. Consequently, the proposed method outperforms the previous state-of-the-art annotation-free instance segmentation methods. In addition, it shows a competitive performance to existing open vocabulary object detection methods without any annotations.

**Strengths:**

## 1. Great motivation and findings.

I strongly agree with the motivation of this work.
Namely, SAM itself may not be suitable for instance segmentation on COCO due to the high recall and low precision rates.
Also, DINO may be suitable for salient object detection, but not for instance segmentation.
Motivated by the limitations of vision foundation models, this paper designed a new pipeline for annotation-free instance segmentation using fine-grained features from CLIP and SAM.

In addition, the proposed classification-first-then-discovery pipeline is convincing in resolving CLIP's misclassifying issue.

## 2. Well-structured and well-written paper

I enjoyed reading this paper because the motivation is clear and understandable, the proposed pipeline is well-explained with proper illustrations.

## 3. Outstanding performance

Combined with CLIP and SAM, the proposed method outperforms the existing methods by a large margin on COCO dataset.
Each proposed component is well-ablated.

**Weaknesses:**

## 1. Limited technical novelty

I feel that leveraging the features on particular middle layers of CLIP and applying K-Means clustering on CLIP's features are not technically novel but well-engineered.

Of course, the proposed annotation-free instance segmentation pipeline is interesting.
This paper seems to be an engineering paper that addresses how to effectively use CLIP and SAM and is not far from an academic paper.

## 2. Behind the outstanding performance.

I think the key feature of the proposed method is how to properly cluster objects in an image (because we regard that SAM can segment anything when bounding box prompts are properly introduced).
At first, I was surprised by the outstanding performance of the proposed method on COCO 2017 dataset.
However, there is a question that the outstanding performance was due to the well-aligned semantic domain between the features of CLIP and the COCO dataset; the clustered object boundary from CLIP may be well-aligned to the ground-truth object region in COCO dataset.
I wonder if the target dataset was human part segmentation or medical segmentation, the CLIP features are still valid for annotation-free instance segmentation.
It would be great if this paper discuss it.

**Questions:**

Due to some concerns in the Weakness part, my initial rating is borderline.
After a discussion with the authors and other reviewers, I will finalize the rating.

**Details Of Ethics Concerns:**

The paper includes ethical considerations and potential negative social impacts, and there don't seem to be any issues to worry about.

---

> ### Author Response · Authors · 2023-11-16
> **Response to Reviewer CLaV**
>
> > W1: Limited technical novelty. I feel that leveraging the features on particular middle layers of CLIP and applying K-Means clustering on CLIP's features are not technically novel but well-engineered. Of course, the proposed annotation-free instance segmentation pipeline is interesting. This paper seems to be an engineering paper that addresses how to effectively use CLIP and SAM and is not far from an academic paper.
> >
>
> Thank you for your thoughtful review and your comments regarding our technical novelty.  We understand your perspective and acknowledge that the application of K-Means clustering to CLIP’s features may seem like an engineered technique when viewed solely from the algorithm execution process. We would like to clarify the technical novelty and contribution that underpin our work as follows:
>
> - **Semantic-Aware Initialization:** we propose a novel semantic-aware initialization technique to leverage semantic activation of CLIP to automatically initialize clustering centers and determine the number of clusters. Without our semantic-aware initialization, K-Means fails to produce general and reliable clustering results for delineating object boundaries. As shown in Figure 5, despite clustering the same intermediate features in CLIP, it is only when utilizing the proposed initialization technique that stable clustering and boundary discovery are achieved. The core motivation behind semantic-aware initialization is that by **initializing certain clustering centers near the outer edges** of the semantic activation map, **boundaries between objects within the activation map** can be clustered and outlined (Please refer to **Appendix L** for a comprehensive explanation). Semantic-aware initialization is a unique technical contribution that sets our work apart from conventional clustering approaches.
> - **Classification-First-Then-Discovery Pipeline:** we propose a classification-first-then-discovery pipeline, which represents a novel paradigm in the field of annotation-free instance segmentation. In the pipeline, we introduce a boundary metric to identify boundaries and transform the instance segmentation challenge into a fragment selection task. This pipeline has not been explored in prior work and offers a fresh perspective on solving the instance segmentation problem.
> - **The Boundary Discovery Property of CLIP**: we are the first to discover that CLIP's specific intermediate layers can delineate object edges, marking an interesting academic finding rather than just an engineering achievement. As discussed in Appendix C.1, we believe that the hierarchical representation structure, distinctive pretraining approach, and training data may be the reasons why CLIP's intermediate features have the potential for instance-level localization. Similar to the salient object discovery capability of the DINO model and the presence of in-context learning and prompting learning in LLM, we hope that this discovery can assist the academic community in better understanding CLIP and exploring its various features and underlying principles of contrastive language-image pre-training.
> - **Clear Motivation and In-depth Analysis with Academic Significance:** Instead of simply combining SAM and CLIP, we conduct a comprehensive analysis of DINO, SAM, and CLIP. This allows us to clearly illustrate our motivation for effectively leveraging foundation models, rather than superficially combining them without thorough consideration. We believe that the effective utilization of CLIP and SAM for annotation-free instance segmentation, as demonstrated in our paper, is a valuable contribution to both the engineering and academic communities.
> - **Simple yet Super Effective and Efficient:** Through our in-depth analysis and well-explored motivation, we were able to propose a method that is both super simple but also super effective. Furthermore, thanks to its simplicity, it is also highly efficient, achieving a 10-fold increase in processing speed compared to previous state-of-the-art unsupervised object discovery methods like CutLER.
>
> > W2: I wonder if the target dataset was human part segmentation or medical segmentation, the CLIP features are still valid for annotation-free instance segmentation.
> >
>
> In addition to the COCO dataset, we apply the same clustering algorithm and exactly the same hyperparameters on the Pascal VOC dataset (as shown in Table 3), and achieve consistently positive results.
>
> Following your suggestion, we also conducted an evaluation of our method on the human part segmentation task, as depicted in the newly added **Figure 12**. We employ the same clustering method and hyperparameters for this evaluation, and the results further demonstrate the generality of our discovery and the robustness of our approach.

---

> ### Comment · Area_Chair_ste7 · 2023-12-04
> **[Important] Response Required to Authors' Rebuttal**
>
> Dear Reviewer CLaV,
>
> As we progress through the review process for ICLR 2024, I would like to remind you of the importance of the rebuttal phase. The authors have submitted their rebuttals, and it is now imperative for you to engage in this critical aspect of the review process.
>
> Please ensure that you read the authors' responses carefully and provide a thoughtful and constructive follow-up. Your feedback is not only essential for the decision-making process but also invaluable for the authors.
>
> Thank you,
>
> ICLR 2024 Area Chair

---

### Official Review · Reviewer_5mwu · 2023-10-30

**Soundness:** 3 good
**Presentation:** 3 good
**Contribution:** 3 good
**Rating:** 6
**Confidence:** 4

**Summary:**

The authors propose a method called ZIP, which
combines CLip and SAM in a novel classification-first-then-discovery pipeline, enabling annotation-free,
 complex-scene-capable, open-vocabulary object detection and instance segmentation.

Zip establishes state-of-the-art performance in various settings, including training-free, self-training,
and label-efficient finetuning.

The overall idea is based on a discovery that CLIP can provide a highly
beneficial and strong instance-level boundary prior in the clustering results of its
particular intermediate layer.

The proposed method combines the CLIP classification score with the SAM localization score using a geometric mean.
This refinement boosts the confidence scores of masks, achieving a slightly higher precision.

**Strengths:**

Good ZSL model proposed- combining CLIP and SAM, and giving semantics meaning to edges.

Large amount of experimental results, exhibiting superiority with SoA, ablation studies etc
provided for benchmark datasets.
Illustrations of results are vivid and eye-catching -= say in Fig. 10.

**Weaknesses:**

The statement used - The devil is in the object boundaries;
is it not in line with a part work, which hypothesized - "all you need are priors"?

Although the  analytics presented in Eqns. (1) - (7) are clear, unable to find much of novelty in them.
eg Cosine similarity, boundary score etc.

What about inference ? How many GPUs are required, and what is the timing then ?
I could not locate that information in document.

A generic opinion:
The recent trend of using a large GPU cluster to implement detectors/segmentors is something of a concern, I believe.
Its like exploiting massive computing power to solve a problem. Either use of a large set of activation/attention heads, \
or for pre-training tasks in self-supervision, meta-learning are the main reasons of using so. What is the limit now ?
The focus is shifting from devising novel learning algorithms to use of larger to mega-clusters.

**Questions:**

Page 2 - the line:
....higher precision yet not satisfaction enough.
may be corrected (English).

You state - "classification-first-then-discovery pipeline"; in page 3:
What if the reverse is done ? Edges in images, as boundaries of objects, may hypothetically lie
on Decision boundaries between binary classes. So discover the edges first may be a better approach, rather than
relying on classification to provide those edges ?

Dino (also Clip?) uses transformers (ViT) , right ?
- is that the reason for the need for large computational resource to solve your problem,
when you combine the two?

---

> ### Author Response · Authors · 2023-11-16
> **Response to Reviewer 5mwu**
>
> Thank you for your detailed review. In order to facilitate a prompt discussion and exchange of ideas, we now address your most critical concerns first. We will respond to all questions and revise the paper as soon as possible. Thanks for your understanding!
>
> > W1: The statement used - The devil is in the object boundaries; is it not in line with a part work, which hypothesized - "all you need are priors"?
> >
>
> Thank you for your valuable suggestions! In fact, "all you need are priors" is among the title candidates we consider. However, we ultimately choose "the devil is in the object boundaries" because we want to highlight that discerning object boundaries is the crucial aspect of how to empower foundation models for annotation-free instance segmentation.
>
> While we recognize that "the devil is in the object boundaries" may not be the optimal choice, we will continue to explore a more suitable title (such as "Unveiling The Object Boundary: Empower Foundation Models Towards Annotation-free Instance Segmentation"). If you have any suggestions or recommendations, please feel free to share them with us. Thank you!
>
> > W2: Although the analytics presented in Eqns. (1) - (7) are clear, unable to find much of novelty in them. eg Cosine similarity, boundary score etc.
> >
>
> Although the implementation of our Zip may seem straightforward, involving simple calculations in Equations (1)-(7), the observations, motivations, technical contributions, and innovations behind Zip are far from trivial.
>
> Moreover, it is precisely due to its simplicity that our method not only demonstrates effectiveness in terms of performance but also remarkable efficiency. When compared to the previous best-performing method, CutLER, under default settings, our algorithm typically delivers a 10 times increase in processing speed.
>
> In addition, our novelies are summary as follows:
>
> - We are the **FIRST to effectively utilize foundation models (CLIP and SAM) for annotation-free instance segmentation.** Such effective utilization is not trivial and cannot be achieved by a naive combination. Simply combining SAM and CLIP results in unsatisfactory performance due to SAM's semantic-unaware issue of generating masks with varying degrees of granularity yet lacking instance-aware discernment (see Appendix B) and CLIP's misclassification issue (see Appendix F). Based on these observations, we propose that the **key to effectively utilizing foundation models for the instance segmentation task is to probe foundation models to delineate the boundaries between individual objects!**
> - We are the **FIRST to discover CLIP's special intermediate layers can outline object edges** through the proposed **novel semantic-aware feature clustering**. By incorporating this discovery, we address challenges in complex scenarios with multiple instances, under annotation-free instance segmentation. To ensure stable and general clustering of object edges, we devise a semantic-aware clustering initialization technique, which is crucial.
> - We are the **FIRST to propose a novel classification-first-then-discovery pipeline** for annotation-free instance segmentation task. Mere identification of boundary information does not inherently and straightforwardly aid instance segmentation without our classification-first-then-discovery pipeline. In the pipeline, we introduce a boundary metric to identify boundaries and transform the instance segmentation challenge into a fragment selection task, a novel approach not explored by previous work.

---

> > ### Author Response · Authors · 2023-11-16
> >
> > > W3: What about inference? How many GPUs are required, and what is the timing then? I could not locate that information in document.
> > >
> >
> > Our Zip is highly efficient, achieving a 10-fold increase in processing speed compared to previous state-of-the-art unsupervised object discovery methods like CutLER, by avoiding computing the eigenvector which is used in the DINO-based method. Since our method is gradient-free, it can run on any GPU with a memory capacity of 12GB. On an A100 with 80GB memory, the average inference time for a single image is less than 2 seconds.
> >
> > > Q2: You state - "classification-first-then-discovery pipeline"; in page 3: What if the reverse is done ? Edges in images, as boundaries of objects, may hypothetically lie on Decision boundaries between binary classes. So discover the edges first may be a better approach, rather than relying on classification to provide those edges ?
> > >
> >
> > The discovery-first-then-classification pipeline could not perform well in detecting objects of the same class that are closely positioned. The edges discovered through the decision boundaries between binary classes typically correspond to outer edges between salient regions, as illustrated by the outer edges of the green "Semantic Clues" in Figure 3. These outer edges may not accurately distinguish the boundaries between individual objects, leading to the segmentation of a cluster of oranges as a single instance rather than recognizing each individual orange.
> >
> > Moreover, inverting the pipeline is an effective approach to validate our classification-first approach. However, directly reversing the pipeline and keeping other components the same is difficult. The reason is that the fragment selection process in the discovery stage requires semantic clues in the classification stage to determine whether to merge groups of fragments into instances. Below, we give two potential reverse pipelines.
> >
> > 1. Building upon the current clustering results, we further cluster on small fragments to form larger connected regions to represent instances. Specifically, we use bottom-up hierarchical clustering to gradually merge fragments into corresponding connected regions based on the similarity between fragments and their adjacent regions. Then, we use CLIP scores to determine the classes and confidence scores.
> > 2. We directly use SAM to segment anything in one image and then use CLIP to classify anything.
> > |  | AP50 | AR100 | Time Speed |
> > | --- | --- | --- | --- |
> > | 1 | 13.5 | 22.1 | 2.3X |
> > | 2 | 16.6 | 51.2 | 1X |
> > | Ours | 27.3 | 32.8 | 4.3X |

---

> > ### Author Response · Authors · 2023-11-21
> >
> > > W4: A generic opinion: The recent trend of using a large GPU cluster to implement detectors/segmentors is something of a concern, I believe. Its like exploiting massive computing power to solve a problem. Either use of a large set of activation/attention heads, or for pre-training tasks in self-supervision, meta-learning are the main reasons of using so. What is the limit now ? The focus is shifting from devising novel learning algorithms to use of larger to mega-clusters.
> > >
> >
> > Training large segmentation models like SAM indeed requires immense computational resources, but once trained, its impact and contributions to the academic community are substantial. The introduction of SAM has led to significant achievements and responses in various fields, including detection, segmentation, medical applications, image editing, and feature extraction. The primary objective of the Zip initiative is to address a challenge unmet by SAM during the pre-training phase: acquiring one billion masks and preparing labels for these masks is nearly impossible. Zip achieves this by fully leveraging the respective capabilities of the foundation models, resulting in a synergistic effect where the combined impact surpasses the sum of individual contributions.
> >
> > > Q3: Dino (also Clip?) uses transformers (ViT) , right ?
> > >
> > > - is that the reason for the need for large computational resource to solve your problem, when you combine the two?
> >
> > Zip can run on any GPU with a memory capacity of 12GB. The computational resource requirements for ZIP are relatively modest.

---

> ### Comment · Area_Chair_ste7 · 2023-12-04
> **[Important] Response Required to Authors' Rebuttal**
>
> Dear Reviewer 5mwu,
>
> As we progress through the review process for ICLR 2024, I would like to remind you of the importance of the rebuttal phase. The authors have submitted their rebuttals, and it is now imperative for you to engage in this critical aspect of the review process.
>
> Please ensure that you read the authors' responses carefully and provide a thoughtful and constructive follow-up. Your feedback is not only essential for the decision-making process but also invaluable for the authors.
>
> Thank you,
>
> ICLR 2024 Area Chair

---

### Official Review · Reviewer_WVZD · 2023-10-30

**Soundness:** 4 excellent
**Presentation:** 3 good
**Contribution:** 4 excellent
**Rating:** 8
**Confidence:** 4

**Summary:**

The authors analyze the weakness of the previous computer vision foundation models. By discovering the instance-level understanding of CLIP, the authors then utlize it to propose a novel training-free method to address the issues by clustering the activation maps from CLIP. Zip demonstrates significant performance improvements compared to the previous state-of-the-art on the COCO dataset across various settings.

**Strengths:**

1. The idea is novel, inspiring and useful.
2. A detailed analysis of the issues in previous works is provided.
3. Significant improvements over previous state-of-the-art on various COCO settings.
4. The authors are aware and thoroughly investigate the weaknesses and limitations of the work.

**Weaknesses:**

1. It is challenging to follow the explanation of how the clustering works. It would be helpful if the authors could include some figures to illustrate this process more clearly.
2. The robustness of the clustering algorithm is concerning when applying to other datasets or settings.

**Questions:**

- Q1. The work is super interesting, but it takes me a long time to understand how the clustering works.
   - What do the colors represented in "Clustering" in Fig.3?
   - How does the clustering algorithm draw the boundaries in Fig.2 and Fig.3?
   - Taking "Semantic Clues" in Fig. 3 as example, what will it look like after applying Eq. 5?  I assume that there will be a single box surrounded all the green parts? How can instances be seperated after that?
   - The authors stated that after obtaining the feature maps for each category, it will then run the clustering algorithm. Please clarify how to run the algorithm and why it will work.
- Q2. How stable is the clustering algorithm? Will the results change drastically if optimal hyperparameters are not found? For example, which intermediate layer should be used? Is there any experiment on how the choice of $K$ will affect the results? Are there any guidelines? Please also provide some insights for Eq.7.
- Q3. Is it appropriate to fully adopt the setting from CutLER? To my understanding, CutLER does not involve any annotation during training. While Zip also does not use annotation from COCO, but it did exploit the language information from CLIP.
- Q4. Is it possible to compare Zip to other open-vocabulary works, such as Openseed[1]? I am aware that Zip is annotation-free, and the authors have stated that it will be a future work to utilize annotations at Section J. However, the work would be more convincing and useful if it is possible to somehow utilize annotations from COCO since the raw performance of Zip is still significantly behind Mask-RCNN.

I belive it is an interesting and promising work. However, the explanation of clustering is not clear enough. I am willing to raise my score if the authors can reasonably address my concerns and questions.

[1] Zhang, Hao, et al. "A simple framework for open-vocabulary segmentation and detection." Proceedings of the IEEE/CVF International Conference on Computer Vision. 2023.

---

> ### Author Response · Authors · 2023-11-16
> **Response to Reviewer WVZD**
>
> Thank you for your detailed review. In order to facilitate a prompt discussion and exchange of ideas, we now address your most critical concerns first. We will respond to all questions and revise the paper as soon as possible. Thanks for your understanding!
>
> > W1+Q1: It is challenging to follow the explanation of how the clustering works. It would be helpful if the authors could include some figures to illustrate this process more clearly.
> >
>
> Thank you for your valuable suggestions! We have added an in-depth explanation of the clustering process in **Appendix L**. Please refer to it for a detailed understanding. The core of the clustering process is our semantic-aware initialization, which leverages the semantic activation map to automatically initialize clustering centers and determine the number of clusters for K-Means.
>
> The core motivation behind semantic-aware initialization is that **by initializing some clustering centers near the outer edges of the semantic activation map, boundaries between objects within the activation map can even be clustered and outlined.** Despite the semantic activation map's inability to differentiate instances with the same classes, its edges continue to signify the boundaries between objects of distinct classes. By initializing some clustering centers near these edges (specifically, the outer edges of the semantic activation map), we can effectively group together the boundaries between objects from **BOTH** the same and different classes into one cluster (i.e., the "boundary" cluster) due to their high feature similarity on the specific intermediate layer of CLIP. This results in the boundaries of objects from the same class also being identified by the "boundary" cluster, thereby enable the distinction of these objects through the "boundary" cluster. As shown in the Figure 5, even though the middle three figures cluster the same intermediate features in CLIP, only clustering with our semantic-aware initialization can deliver stable clustering and boundary discovery.
>
> > W2: The robustness of the clustering algorithm is concerning when applying to other datasets or settings.
> >
>
> In addition to the COCO dataset, we apply the same clustering algorithm and exactly the same hyperparameters on the Pascal VOC dataset (as shown in Table 3), and achieve consistently positive results.
>
> We also conduct an evaluation of our method on the human part segmentation task, as depicted in the newly added **Figure 12**. We employ the same clustering method and hyperparameters for this evaluation, and the results further demonstrate the robustness of our approach.
>
> > Q2: How stable is the clustering algorithm? Will the results change drastically if optimal hyperparameters are not found? For example, which intermediate layer should be used? Is there any experiment on how the choice of  K will affect the results? Are there any guidelines? Please also provide some insights for Eq.7.
> >
>
> Thanks for the valuable feedback. We agree that hyperparameter tuning and large search space are challenging in most unsupervised methods. As a reference, the previous SOTA CutLER has nine hyperparameters.
>
> Essentially, our clustering algorithm has two hyperparameters. Most importantly, these hyperparameters are relatively independent, easy to determine, and consistent across different datasets and experimental settings.
>
> - The setting of threshold $\tau$=0.7 in Equ (5) is consistent across different datasets. The $\tau$ determines whether the pixel blocks are activated given a category, i.e., the threshold to judge whether pixel blocks are the category's foreground or background. It is an unavoidable hyper-parameter to binarize a similar/confidence map to a binary mask in the previous unsupervised object detection methods. Fortunately, our approach only needs a coarse binary mask to help initialize cluster centers instead of a precise one for final prediction, increasing the robustness to $\tau$ and setting $\tau$ consistently over different tasks and datasets.
> - The threshold in Equ (7) is set to 0.15 in all the experiments. Equ (7) roughly estimates the number of clusters based on estimating the number of object categories in single images. In reality, our method only needs to set the number of clusters within a certain range (15~30) to work effectively, thanks to our reformulation instance segmentation by fragment selection.
> Our method is even robust to over-segmentation caused by a slightly larger number of clusters (as long as it is not too small) because our subsequent fragment selection (Sec 3.4) can regroup over-segmented fragments into an instance.
> | Pascal VOC | AP50 | AR100 |
> | --- | --- | --- |
> | K=15 | 21.9 | 25.5 |
> | Ours | 22.4 | 28.4 |
> | K=30 | 21.1 | 26.6 |

---

> > ### Author Response · Authors · 2023-11-21
> >
> > > Q3. Is it appropriate to fully adopt the setting from CutLER? To my understanding, CutLER does not involve any annotation during training. While Zip also does not use annotation from COCO, but it did exploit the language information from CLIP.
> > >
> >
> > It's crucial to note that CutLER utilizes the object-centric dataset ImageNet for generating pseudo labels and self-training.  In addition, we involve several attempts to ensure a fair comparison. 1. In the class-aware setting, it's natural for unsupervised methods to leverage CLIP for recognition, and our use of CLIP does not introduce additional overhead. 2. In Table 1, we mainly highlight the significant improvement compared to the original SAM+CLIP approach.
> >
> > > Q4. Is it possible to compare Zip to other open-vocabulary works, such as Openseed[1]? I am aware that Zip is annotation-free, and the authors have stated that it will be a future work to utilize annotations at Section J. However, the work would be more convincing and useful if it is possible to somehow utilize annotations from COCO since the raw performance of Zip is still significantly behind Mask-RCNN.
> > >
> >
> > Indeed, exploring the potential of Zip in labeled scenarios is a promising avenue for future research. In our main text, we delved into what Zip can achieve in labeled conditions (see Section 4.3 on label-efficient experiment). Our method, when trained with only 1% of the data, surpasses the performance of the previous state-of-the-art unsupervised method MOCO-V2 by over 5 percentage points.
> >
> > OpenSeed intends to jointly train on the extensive datasets of COCO and Object365. In order to facilitate a meaningful comparison with OpenSeed, it is conceivable that we may also need to augment our training data to elevate its volume. Although it goes beyond the scope of this discussion, we would be delighted to share any new results on this topic if available.

---

> ### Comment · Reviewer_WVZD · 2023-11-21
>
> Thanks for the detailed replies. I have no more questions and willing to raise my score.
> Here are some minor issues.
> 1. In Appendix L, should Figure 13 be Figure 11 instead?
> 2. I assume the outer edge at Figure 11(d) should be the contour of the oranges not the brown box of the images? It is somewhat confusing.
> 3. The discussion of which layer to use in different scenarios is still lacking.
> 4. The explanations of semantic aware initialization should be intergrated for the final version. I can't understand how it work without looking at the appendix.

---

> > ### Author Response · Authors · 2023-11-22
> >
> > We thank the reviewer for the quick reply and consideration!
> >
> > 1. We have rectified the citation issues in the Appendix L. Appreciate your attention to this matter!
> > 2. We rename Figure 11(d) from “outer edge” to “Contour of the oranges” and remove the brown box of the image.
> > 3. In Figure 5, We provide visualization results for various layers. As we only consistently observe the unique phenomenon of CLIP at a specific layer (conv4_x), we adopt CLIP feature from conv4_x in all experiment. Thanks for you kind suggestion, we will include quantitative results regarding the middle layer.
> > 4. We will restructure the organization of the article, incorporating explanations into the main body.
> >
> > Your detailed review and feedback are valuable to us in further shaping our work. Once again, we express our gratitude for your time and appreciation!

---

> ### Comment · Area_Chair_ste7 · 2023-12-04
> **[Important] Response Required to Authors' Rebuttal**
>
> Dear Reviewer WVZD,
>
> As we progress through the review process for ICLR 2024, I would like to remind you of the importance of the rebuttal phase. The authors have submitted their rebuttals, and it is now imperative for you to engage in this critical aspect of the review process.
>
> Please ensure that you read the authors' responses carefully and provide a thoughtful and constructive follow-up. Your feedback is not only essential for the decision-making process but also invaluable for the authors.
>
> Thank you,
>
> ICLR 2024 Area Chair

---

### Official Review · Reviewer_g6s6 · 2023-10-31

**Soundness:** 3 good
**Presentation:** 3 good
**Contribution:** 3 good
**Rating:** 6
**Confidence:** 4

**Summary:**

This paper introduces a novel approach to annotation-free instance segmentation. The authors identify that a specific middle-level feature in CLIP encapsulates instance-level boundary information. Leveraging this insight, they develop a pipeline named "Zip" for annotation-free instance segmentation. The paper also pioneers a "classification-first-then-discovery" paradigm. Through extensive experiments, the proposed method significantly outperforms its baselines.

**Strengths:**

The utilization of CLIP's middle-level feature to generate instance proposals presents a compelling and novel approach. This stands in stark contrast to traditional RPN-based or embedding-based proposal generation methods, offering a fresh perspective in the realm of annotation-free instance segmentation. It also provide comprehensive experiment results to prove its leading performance.

**Weaknesses:**

1. **Figure 1's Layout**: The sequencing in Figure 1 is somewhat perplexing. A more intuitive order, such as DINO, Prev. SOTA, SAM, Clustering, and then Ours, might enhance clarity. Additionally, specifying the exact method when referencing "Prev. SOTA" would provide better context to the readers.

2. **Figure 2's Clarity**: The visualization in Figure 2 appears cluttered, making it challenging to discern the boundaries in images labeled A through D. Simplifying or enhancing the contrast could make the distinctions more evident.

3. **Understanding Figure 3**: Figure 3 is somewhat intricate, especially when trying to comprehend the roles of "activation" and "boundary." A more detailed caption or a supplementary explanation might aid in understanding.

4. **Discussion on DINO and SAM**: While it's essential to provide context, the extensive space dedicated to discussing the limitations of DINO and SAM might be excessive. Streamlining this section could make the paper more concise and focused on the primary contributions.

**Questions:**

1. **Semantic-Aware Initialization**: Could you provide a more in-depth explanation of the Semantic-Aware Initialization mentioned on page 17? Specifically, I'm unclear about how the parameters for K-means clustering are determined.

2. **Visualization in Figure 2**: The differentiation between gray and orange in Figure 2 could be clearer. Enhancing the contrast by making the background color more transparent might help in better distinguishing the two.

3. **Ambiguous Object Representation**: How does the middle-level clustering handle images where the concept of an object is ambiguous? For instance, in scenarios like a box with a person's image on one side or stacked blocks. In such cases, it's debatable whether the person's image should be considered a separate instance or if each block in the stack should be individually segmented.

4. **Handling Occlusions**: How does your method address situations where an object is partially occluded and appears as two separate parts? For example, if a dog is behind a tree, would your system recognize both parts as belonging to the same object?

5. **Locating Middle-Level Features**: I'm intrigued about the process of pinpointing the middle-level feature. Is the method you've employed reproducible across different training weights? For instance, would the results be consistent if you were to use the openCLIP model (or the official CLIP model, if you've already utilized openCLIP)?

6. **Exploring DINO/DINOv2 Features**: Have you considered using the middle-level features from DINO or DINOv2? Given that MaskCut can discern between closely packed objects using DINO features, I wonder if DINO/DINOv2 might also possess a middle-level feature adept at boundary extraction.

7. **Resolution of Middle-Level Feature**: The resolution of your middle-level feature appears to be quite high. Could you specify the image input size and the dimensions of your middle-level feature?

8. **Self-Training Visualization**: It might be beneficial to include a comprehensive visual representation of the self-training process. Relying solely on textual differences compared to CutLER and directing readers to refer to CutLER might not be the most user-friendly approach.

---

> ### Author Response · Authors · 2023-11-16
> **Response to Reviewer g6s6**
>
> Thank you for your detailed review. In order to facilitate a prompt discussion and exchange of ideas, we now address your most critical concerns first. We will respond to all questions and revise the paper as soon as possible. Thanks for your understanding!
>
> > W1+W2+Q2:  Figure 1's Layout, Figure 2's Clarity, and Visualization in Figure 2
> >
>
> Thanks for your suggestion. We will improve the alignment in Figure 1 and enhance the clarity of boundaries in Figure 2 to enhance visualization.
>
> > W3: Understanding Figure 3.
> >
>
> Thank you for pointing this out. We will revise the caption as follows: ZIP follows a classification-first-then-discovery approach, consisting of four steps: 1) Classification first to obtain semantic clues provided by CLIP, where the semantic clues indicate the approximate activation regions of potential objects. 2) Clustering on CLIP's features at a specific intermediate layer to discover object boundaries with the aid of our semantic-aware initialization. The semantic-aware initialization leverages semantic activation to automatically initialize clustering centers and determine the number of clusters. 3) Localization of individual objects by regrouping dispersed clustered fragments that have the same semantics, all while adhering to the detected boundaries. 4) Prompting SAM for precise masks for each individual object.
>
> > W4: Discussion on DINO and SAM.
> >
>
> Thank you for your suggestion. We will relocate some of the discussions about DINO and SAM to the supplementary materials and place a stronger emphasis on our primary contributions in the introduction.
>
> > Q1: **Semantic-Aware Initialization**: Could you provide a more in-depth explanation of the Semantic-Aware Initialization mentioned on page 17?
> >
>
> Thanks for pointing this out. We have added an in-depth explanation of the semantic-aware initialization in **Appendix L**. Please refer to it for a detailed understanding.
>
> The core motivation behind semantic-aware initialization is that **by initializing some clustering centers near the outer edges of the semantic activation map, boundaries between objects within the activation map can even be clustered and outlined.** Despite the semantic activation map's inability to differentiate instances with the same classes, its edges continue to signify the boundaries between objects of distinct classes. By initializing some clustering centers near these edges (specifically, the outer edges of the semantic activation map), we can effectively group the boundaries between objects from **BOTH** the same and different classes into one cluster (i.e., the "boundary" cluster) due to their high feature similarity on the specific intermediate layer of CLIP. This results in the boundaries of objects from the same class also being identified by the "boundary" cluster, thereby enabling the distinction of these objects through the "boundary" cluster. As shown in Figure 5, even though the middle three figures cluster the same intermediate features in CLIP, only clustering with our semantic-aware initialization can deliver stable clustering and boundary discovery.

---

> > ### Author Response · Authors · 2023-11-21
> >
> > > Q2: **Visualization in Figure 2**: The differentiation between gray and orange in Figure 2 could be clearer. Enhancing the contrast by making the background color more transparent might help in better distinguishing the two.
> > >
> >
> > We appreciate your guidance, and we have refined Figure 2 accordingly.
> >
> > > Q3: **Ambiguous Object Representation**: How does the middle-level clustering handle images where the concept of an object is ambiguous? For instance, in scenarios like a box with a person's image on one side or stacked blocks. In such cases, it's debatable whether the person's image should be considered a separate instance or if each block in the stack should be individually segmented.
> > >
> >
> > It's indeed an interesting question. We acknowledge that handling the concept of an object, especially in the unsupervised domain, is challenging. However, our method has two advantages compared to the methods mentioned in the literature. First, we identified the case you mentioned, "people in tv," in the VOC dataset and provided clustering results in Figure 13(d). We observe that our clustering method avoids losing fragments of people. Thanks to our **semantic-aware initialization** clustering approach, coupled with the utilization of semantic activation maps, both human figures and television entities can be discerned and segmented (see **Appendix L**), our method can successfully segment both people and TVs, demonstrating its capability regardless of the object being segmented.
> >
> > > Q4: **Handling Occlusions**: How does your method address situations where an object is partially occluded and appears as two separate parts? For example, if a dog is behind a tree, would your system recognize both parts as belonging to the same object?
> > >
> >
> > In our current approach, it is less likely for occluded objects to be merged together because we assume that objects are connected, and we run a connected components algorithm to eliminate some noise outliers. However, addressing occlusion is indeed a potential and interesting avenue for future research.
> >
> > > Q5: **Locating Middle-Level Features**: I'm intrigued by the process of pinpointing the middle-level feature. Is the method you've employed reproducible across different training weights? For instance, would the results be consistent if you were to use the openCLIP model (or the official CLIP model, if you've already utilized openCLIP)?
> > >
> >
> > We adopt official CLIP implement and weights as default. Following your suggestion, we experimented with various models in Open-CLIP, including RN50*64, ViT-bigG-14, convnext_xlarge, and ViT-B-16-quickgelu, along with diverse training datasets (DataComp-1B, CommonPool-L). Yet, there was no consistent occurrence of boundary phenomena. It appears that the training data employed by official CLIP possesses distinctive characteristics.
> >
> > > Q6: **Exploring DINO/DINOv2 Features**: Have you considered using the middle-level features from DINO or DINOv2? Given that MaskCut can discern between closely packed objects using DINO features, I wonder if DINO/DINOv2 might also possess a middle-level feature adept at boundary extraction.
> > >
> >
> > We adopt the same cluster algorithm and setting as CLIP on the DINOv2 middle-layer feature in Figure 13. The clustering outcomes of DINO appear notably smoother. We haven't observed the presence of boundary phenomena akin to those found within CLIP.
> >
> > > Q7: **Resolution of Middle-Level Feature**: The resolution of your middle-level feature appears to be quite high. Could you specify the image input size and the dimensions of your middle-level feature?
> > >
> >
> > Our image size is 2048*2048 (see implementation), and the middle-level feature’s shape is 64*64*1024. Our zip can run on any GPU with a memory capacity of 12GB.
> >
> > > Q8. **Self-Training Visualization**: It might be beneficial to include a comprehensive visual representation of the self-training process. Relying solely on textual differences compared to CutLER and directing readers to refer to CutLER might not be the most user-friendly approach.
> > >
> >
> > Thanks to your suggestion, Our self-training employs the Cascade Mask R-CNN with ResNet-50 backbone. Our batch size is 16 and self-train for 160K iteration. For the optimizer we leverage SGD with an initial learning rate of 0.005 which is diminished by a factor of 5 after 80K iterations, a weight decay of 5e-5, and a momentum of 0.9. We include all self-training details in Appendix A.3

---

> > > ### Comment · Reviewer_g6s6 · 2023-11-23
> > >
> > > Overall I think this observation proposed in this paper is interesting, but its significance is undermined by two major issues: Firstly, the figures are not clear enough to effectively demonstrate the detailed pipeline. Secondly, the observation's exclusivity to official CLIP and its non-replicability in openCLIP limit its broader impact, as it leaves unanswered questions about the underlying properties causing these results

---

> > > > ### Author Response · Authors · 2023-11-23
> > > >
> > > > We appreciate the reviewer for recognizing our observation is interesting and for the reviewer's engagement in assessing our work!
> > > >
> > > > - For the figures, we add explanatory Figure 11 in Appendix L to clarify the clustering process and segment selection process. We will include this figure in the main body and also refine the figures in the paper. Thank you for your invaluable comments!
> > > > - For openCLIP and official CLIP, there are different observations, and we believe the most significant reason for this is **the use of different global pooling layers.**
> > > >     - In official CLIP, attention pooling with the [CLS] token is used to obtain global image features, offering attention maps for free and allowing features with low attention scores to focus on aspects beyond "category" and semantics, such as boundaries. However, openCLIP uses global average pooling to acquire global image features, emphasizing that features from different spatial positions pay more attention to aspects related to "category" and semantics. The importance of different attention pooling mechanisms in identifying boundaries is observed and discussed in the paper [Keep It SimPool: Who Said Supervised Transformers Suffer from Attention Deficit? (ICCV 2023)].
> > > >     - To further validate the above idea, we check different openCLIP models and found that only the "modified RN101" model in openCLIP uses the attention pooling layer following the official CLIP. All other openCLIP models utilize the original global average pooling layer. The newly added Figure 14 shows that only "modified RN101" openCLIP model with **attention pooling layer** (Figure 14.a) can identify object boundaries, while other openCLIP models using **global average pooling layers** fail to outline boundaries.
> > > >
> > > > We thank the reviewer again for the response and further consideration!

---

> ### Comment · Area_Chair_ste7 · 2023-12-04
> **[Important] Response Required to Authors' Rebuttal**
>
> Dear Reviewer g6s6,
>
> As we progress through the review process for ICLR 2024, I would like to remind you of the importance of the rebuttal phase. The authors have submitted their rebuttals, and it is now imperative for you to engage in this critical aspect of the review process.
>
> Please ensure that you read the authors' responses carefully and provide a thoughtful and constructive follow-up. Your feedback is not only essential for the decision-making process but also invaluable for the authors.
>
> Thank you,
>
> ICLR 2024 Area Chair

---

### Official Review · Reviewer_11pM · 2023-10-31

**Soundness:** 2 fair
**Presentation:** 2 fair
**Contribution:** 2 fair
**Rating:** 5
**Confidence:** 4

**Summary:**

Authors propose an annotation free instance segmentation through combining CLIP and SAM, where the authors claim that CLIP has a better capability of obtaining the boundaries. They evaluate their method on COCO and PASCAL VOC datasets. Their framework named Zip outperforms some of the unsupervised annotation free methods from SOA.

**Strengths:**

- interesting results in Table 1 and Table 2 improving over the compared methods
- Interesting direction to work on establishing methods for annotation free instance segmentation

**Weaknesses:**

- Weak novelty as the method is simply utilizing other foundation models without proposing anything specific to the instance segmentation task. The clustering techniques are not really showing anything novel in its formulation. Novelty is more in the pipeline.

- There are no quantitative results to support their claim that DINO is worse than CLIP in identifying the boundaries as far as I have seen but if there are please clarify in the response. Also is CLIP better than Up-DetR or better than DINO, DINOv2 in identifying boundaries. Meaning if they apply same clustering and everything on these models' features how would it fair against CLIP.

- Up-DetR is not clearly stated in the related work and I am wondering if the authors have inspected its use
Dai, Zhigang, et al. "Up-detr: Unsupervised pre-training for object detection with transformers." Proceedings of the IEEE/CVF conference on computer vision and pattern recognition. 2021.

- Equation 3 and the method is not quite clear for example when referring to performing inner product of matrices. But there is no mention of the matrices shapes which can help clarify ambiguities in the mathematical formulation.

- The method is quite heuristic depending on hyper parameters used in Eq. 4 theta_1, theta_2.

- Their results are quite low in Table 1 compared to for example SAM when coupled with ViTDet as reported in original SAM paper. It has to be highlighted on why the previous results weren’t compared to. For example if it was not fully annotation free in SAM paper, please detail that in the paper then.

- Typos needs to be fixed e.g. “classification-frist-then-discovery “

**Questions:**

- Table 3 analysis on the architecture its not clear how the AP50 climbed to 44.9%, I am not sure how did this happen and is this still annotation free? Why it is not the final results compared in Table 1.

- F.g 6 C.1 its not clear what’s the x-axis?

- How did you retrieve CutLER results in Table 1? Why is there one class aware and class agnostic?

---

> ### Author Response · Authors · 2023-11-16
> **Response to Reviewer 11pM**
>
> Thank you for your detailed review. In order to facilitate a prompt discussion and exchange of ideas, we now address your most critical concerns first. We will respond to all concerns and revise the paper as soon as possible. Thanks for your understanding!
>
> > W1: Weak novelty as the method is simply utilizing other foundation models without proposing anything specific to the instance segmentation task. The clustering techniques are not really showing anything novel in its formulation. Novelty is more in the pipeline.
> >
>
> Thank you for your rigorous consideration.
>
> First, we attempt to directly address the reviewer's question regarding the novelty of our utilization of the foundation model and clustering technology.
>
> - **Our method goes far beyond the simple utilization of foundation models without proposing anything specific to the instance segmentation task.** On the contrary, one of our key contributions and novelties lies in how effectively we probe foundation models for annotation-free instance segmentation.
>     - SAM itself is not suitable for instance segmentation on COCO, exhibiting high recall but low precision rates primarily because of its semantic-unaware and mostly edge-oriented approach. Also, DINO may be suitable for salient object detection, but not for instance segmentation. Please refer to the second paragraph of the Introduction and Appendix B.
>     - The simple utilization of CLIP+SAM results in a less-than-ideal performance (see Table 1), highlighting the non-trivial nature of utilizing CLIP and SAM for annotation-free instance segmentation.
>         - CLIP+SAM: CLIP first generates a semantic activation map, and then SAM directly selects points from the activation map to prompt SAM in instance segmentation. This simple utilization results in clustered objects of the same class being unable to be distinguished as individual instances, as the semantic activation map only identifies these objects as a whole.
>         - SAM+CLIP: SAM first produces numerous valid segmentation masks, and CLIP then recognizes these proposals. However, this simple combination yields poor performance due to the SAM's semantic-unaware issue of generating masks with varying degrees of granularity yet lacking instance-aware discernment (see Appendix B) and CLIP’s misclassification issue (see Appendix F).
>
>     Based on these observations and motivations, we propose that the **key to effectively utilizing foundation models for the instance segmentation task is to probe foundation models to delineate the boundaries between individual objects!**
>
> - **For clustering, we propose a novel semantic-aware initialization** technique to automatically initialize clustering centers and determine the number of clusters. **Without** our novel semantic-aware initialization, clustering techniques (such as the K-means we used in our clustering) **CANNOT** generate general and reliable clustering results for outlining object boundaries. As shown in Figure 5, even though the middle three figures cluster the same intermediate features in CLIP, only clustering with the proposed initialization technique can deliver stable clustering and boundary discovery.
>     - The core motivation behind semantic-aware initialization is that **by initializing some clustering centers near the outer edges of the semantic activation map, boundaries between objects within the activation map can even be clustered and outlined.** Despite the semantic activation map's inability to differentiate instances with the same classes, its edges continue to signify the boundaries between objects of distinct classes. By initializing some clustering centers near these edges (specifically, the outer edges of the semantic activation map), we can effectively group together the boundaries between objects from **BOTH** the same and different classes into one cluster (i.e., the "boundary" cluster) due to their high feature similarity on the specific intermediate layer of CLIP. This results in the boundaries of objects from the same class also being identified by the "boundary" cluster, thereby enabling the distinction of these objects through the "boundary" cluster.
>     - We have added a detailed explanation of the clustering process and motivation in **Appendix L**. Please refer to it for a comprehensive understanding.

---

> ### Author Response · Authors · 2023-11-16
>
> ---
>
> Second, we emphasize our novelty from a comprehensive perspective.
>
> - We are the **FIRST to effectively utilize foundation models (CLIP and SAM) for annotation-free instance segmentation**. Such effective utilization is not trivial and cannot be achieved by simply combining foundation models.
> - We are the **FIRST to discover CLIP's special intermediate layers can outline object edges through the proposed semantic-aware feature clustering**. By incorporating this discovery, we address challenges in complex scenarios with multiple instances, under annotation-free instance segmentation. To ensure stable and general clustering of object edges, we devise a semantic-aware clustering initialization technique, which is crucial.
> - We are the **FIRST to propose the classification-first-then-discovery pipeline** for annotation-free instance segmentation task. Mere identification of boundary information does not inherently and straightforwardly aid instance segmentation without our classification-first-then-discovery pipeline. In the pipeline, we introduce a boundary metric to identify boundaries and transform the instance segmentation challenge into a fragment selection task, a novel approach not explored by previous work.
>
> > W2: there are no quantitative results to support their claim that DINO is worse than CLIP in identifying the boundaries as far as I have seen but if there are please clarify in the response. Also is CLIP better than Up-DetR or better than DINO, DINOv2 in identifying boundaries. Meaning if they apply same clustering and everything on these models' features how would it fair against CLIP.
> >
>
> Thank you for your suggestions. However, we cannot **apply the same clustering and everything on features** from DINO, DINOv2, or the Up-DetR models due to the following reasons:
>
> - Our clustering relies on the proposed semantic-aware initialization (see **Appendix L**), which uses CLIP's semantic activation map to effectively initialize clustering centers and determine the number of clusters. However, DINO, DINOv2, or the Up-DetR models cannot produce such a semantic activation map for a class. As a solution, we utilize CLIP to acquire the semantic activation map and subsequently apply the same clustering method on DINOv2. The results are depicted in Figure 13, clearly illustrating that CLIP's clustering results can identify edges, such as the object boundaries of the "person", a task that DINOv2 does not work well.
> - We reformulate the instance segmentation task as a fragment selection task, and the progress of fragment selection also depends on the semantic activation map (see Equ.3). Therefore, we cannot leverage DINO, DINOv2, or the Up-DetR models to accomplish fragment selection.
>
> DINO and DINOv2 can detect salient objects in object-centric images from ImageNet. However, they are not suitable for localizing multiple objects in more complex scenes from COCO. In object detection scenes, DINO and DINOv2 struggle to distinguish between instance-level objects and often miss potential objects within the background.
>
> > W3: Up-DetR is not clearly stated in the related work
> >
>
> Up-DetR is an unsupervised **pre-training** method for object detection. Without annotated data for object detection, Up-DetR cannot solely perform annotation-free object detection or instance segmentation. Up-DetR requires fine-tuning on an object detection dataset after unsupervised pre-training to enable effective object detection. In contrast, our ZIP is entirely annotation-free and does not rely on object detection or instance segmentation annotations.

---

> > ### Author Response · Authors · 2023-11-21
> >
> > > W4: Equation 3 and the method is not quite clear for example when referring to performing inner product of matrices. But there is no mention of the matrices shapes which can help clarify ambiguities in the mathematical formulation.
> > >
> >
> >  The matrices shape for $C_k$, $S_T$ and $1(X_K^T)$ is all H*W*1. And thanks for your suggestion, we'll make sure to specify the shape of all matrices.
> >
> > > W5: The method is quite heuristic depending on hyper parameters used in Eq. 4 theta_1, theta_2.
> > >
> >
> > The parameter $\theta_1$ serves as a threshold to judge whether pixel blocks are the category’s foreground or background and $\theta_2$ serves as a threshold to judge whether a fragment belongs to a boundary. We visualize the boundary scores of fragments sampled from COCO dataset in Fig6. We observe that boundary pixels exhibit notably high boundary scores>5,  In contrast, typical foreground objects tend to have scores below 3. Hence, $\theta_2$ can be adjusted within a broader range without affecting performance. Below is the ablation table for the impact of $\theta_2$
> >
> > | Pascal VOC | AP50 | AR100 |
> > | --- | --- | --- |
> > | $\theta_2$=2.5 | 21.5 | 25.5 |
> > | $\theta_2$=2.7 | 22.2 | 27.8 |
> > | Ours ($\theta_2$=3) | 22.4 | 28.4 |
> > | $\theta_2$=3.5 | 19.4 | 23.4 |
> >
> > Thanks for the valuable feedback. We agree that hyper-parameter is important in most unsupervised methods. As a reference, the previous unsupervised object detection SOTA CutLER, has nine hyperparameters. Most importantly, these hyperparameters in our Zip are relatively independent, easy to be determined, and consistent across different datasets and experimental settings.
> >
> > > W6: Their results are quite low in Table 1 compared to for example SAM when coupled with ViTDet as reported in original SAM paper. It has to be highlighted on why the previous results weren’t compared to. For example if it was not fully annotation free in SAM paper, please detail that in the paper then.
> > >
> >
> > Thank you for your valuable suggestions! The original SAM paper used boxes generated by a fully-supervised ViTDet on COCO as prompts, while our approach does not rely on any COCO annotations. We have added emphasis on the description of the setting in the main text.
> >
> > > Q1: Table 3 analysis on the architecture its not clear how the AP50 climbed to 44.9%, I am not sure how did this happen and is this still annotation free? Why it is not the final results compared in Table 1.
> > >
> >
> > The results in Table 3 are on the Pascal VOC validation set, while Table 1 is on COCO. We chose different datasets to showcase more results and demonstrate the robustness of the methods.
> >
> > > Q2: F.g 6 C.1 its not clear what’s the x-axis?
> > >
> >
> >  It’s ``boundary score'' as defined in equ3
> >
> > > Q3: How did you retrieve CutLER results in Table 1? Why is there one class aware and class agnostic?
> > >
> >
> > Conventional object detection requires both object localization and classification. Previous methods like CutLER lac ked the ability to classify detected objects. Thanks to CLIP, our approach inherently has the capability to determine the category of detected objects without adding extra overhead. To align more closely with conventional object detection, we use the same CLIP model to classify the categories of all proposals generated by previous methods. The specific code for category classification is provided in Appendix A.2.

---

### Author Response · Authors · 2023-11-21

We are grateful for the insightful and inspiring feedback from reviewers. We appreciate the recognition and endorsement of our proposed approach, considering our findings as novel (****g6s6****, ****WVZD****), compelling (****g6s6****), and inspiring (****WVZD****), our motivation as great(****CLaV****), and our performance as strong (****11pM, g6s6, WVZD, 5mwu, CLaV****). We eagerly anticipate any further discussions or inquiries that the reviewers may wish to engage in with us. Throughout our current revision and discussions, we have delved into a comprehensive explanation of how the clustering process is conducted and elucidated our technical contributions.

1. We are the first **to discover CLIP's special intermediate layers can outline object edges.**
2. Inspired by our findings, we propose a **novel semantic-aware initialization technique** to leverage semantic activation of CLIP to automatically initialize clustering centers and determine the number of clusters. Without our semantic-aware initialization, K-Means fails to produce general and reliable clustering results for delineating object boundaries.
3. We are the first **to propose a novel classification-first-then-discovery pipeline** for annotation-free instance segmentation task which **effectively** combine two foundation model SAM and CLIP.
4. Our method is **Simple yet Super Effective and Efficient,** which outperforms previous methods in both performance (+9.7\% AP on class-aware COCO) and efficiency (10X faster).

Specifically, we have implemented the following improvements:

1. Appendix L: Comprehensive explanation of how the clustering process is conducted
2. Appendix M: Applying our clustering methodology on human part images, we demonstrate the robustness of our clustering approach in the presence of CLIP features.
3. Appendix N,I: We expand the quantitative and qualitative analysis between DINO and CLIP features, strengthing our conclusions.
4. Figure 1,2,3: Improved image arrangement, readability, and captions to facilitate better understanding for the readers.
5. Writing: Correcte typos and provide additional explanations for the formulas.

---

### Meta-Review · Area_Chair_ste7 · 2023-12-09

**Metareview:**

The paper under consideration introduces an approach to annotation-free instance segmentation by leveraging the features of specific intermediate layers of CLIP and proposing a novel semantic-aware initialization technique. The average review score stands at a borderline 6, which leads to a recommendation for acceptance, although it remains close to the threshold where rejection could be justifiable. The authors have diligently and comprehensively responded to the concerns raised by the reviewers, particularly addressing the novelty of their approach and the effectiveness of their method in the context of instance segmentation.

**Justification For Why Not Higher Score:**

The paper score is 6, which is at the boundary of the acceptance line. It can be accepted as a poster, but there still remain a few issues to be addressed.

**Justification For Why Not Lower Score:**

The authors' comprehensive response to the reviews, coupled with expanded experiments and improvements in performance and efficiency, substantiates the paper's contribution.

---

### Decision · Program_Chairs · 2024-01-16

Accept (poster)